# Asymmetric cortical projections to striatal direct and indirect pathways distinctly control actions

**Jason R Klug[1†], Xunyi Yan[1,2†], Hilary Hoffman[1], Max D Engelhardt[1], Fumitaka Osakada[3], Edward M Callaway[3], Xin Jin[1,2,4*]**

[1]Molecular Neurobiology Laboratory, The Salk Institute for Biological Studies, La Jolla, United States; [2]New Cornerstone Science Laboratory, Center for Motor Control and Disease, Key Laboratory of Brain Functional Genomics, East China Normal University, Shanghai, China; [3]Systems Neurobiology Laboratories, The Salk Institute for Biological Studies, La Jolla, United States; [4]NYU–ECNU Institute of Brain and Cognitive Science, New York University Shanghai, Shanghai, China

**\*For correspondence:**
xjin@bio.ecnu.edu.cn

[†]These authors contributed equally to this work

**Competing interest:** The authors declare that no competing interests exist.

## eLife Assessment

This manuscript presents an **important** finding that D1- and D2-striatal neurons receive distinct cortical inputs, offering key insights into corticostriatal function. For instance, in the context of striatal-dependent learning, this distinction is highly informative for interpreting synaptic physiology data, particularly when inputs to one neuron subtype may change independently of the other. The strength of the evidence is **solid**, with anatomical and electrophysiological findings aligning well with results from optogenetic and behavioral studies. The study would be of interest to neuroscientists studying basal ganglia circuits in health and disease.

**Abstract** The striatal direct and indirect pathways constitute the core for basal ganglia function in action control. Although both striatal D1- and D2-spiny projection neurons (SPNs) receive excitatory inputs from the cerebral cortex, whether or not they share inputs from the same cortical neurons, and how pathway-specific corticostriatal projections control behavior remain largely unknown. Here using a G-deleted rabies system in mice, we found that more than two-thirds of excitatory inputs to D2-SPNs also target D1-SPNs, while only one-third do so vice versa. Optogenetic stimulation of striatal D1- vs. D2-SPN-projecting cortical neurons differently regulate locomotion, reinforcement learning, and sequence behavior, implying the functional dichotomy of pathway-specific corticostriatal subcircuits. These results reveal the partially segregated yet asymmetrically overlapping cortical projections on striatal D1- vs. D2-SPNs, and that the pathway-specific corticostriatal subcircuits distinctly control behavior. It has important implications in a wide range of neurological and psychiatric diseases affecting cortico-basal ganglia circuitry.

## Introduction

The corticostriatal circuits are critically involved in sensory, cognition, and the learning and control of actions (*Aoki et al., 2019*; *Graybiel, 1998*; *Haber, 2016*; *Hikosaka et al., 1998*; *Jin and Costa, 2010*; *Jin and Costa, 2015*; *Kupferschmidt et al., 2017*; *Stephenson-Jones et al., 2011*; *Tanji, 2001*; *Yin and Knowlton, 2006*). Dysfunctional corticostriatal circuitry has been implicated in numerous neurological and psychiatric diseases (*Shepherd, 2013*), including Parkinson's (*Redgrave et al., 2010*), autism (*Monteiro and Feng, 2017*), and obsessive-compulsive disorder (*Dalley and Robbins, 2017*).

The striatal direct and indirect pathways, made up of D1- vs. D2-expressing spiny projection neurons (SPNs) respectively, constitute the core components for basal ganglia functions in relation to action learning and movement control (*Albin et al., 1989*; *DeLong, 1990*; *Gerfen et al., 1990*). Numerous studies have suggested that the two pathways play distinct yet complementary roles in controlling actions (*Cui et al., 2013*; *Geddes et al., 2018*; *Hikosaka et al., 2019*; *Hikosaka et al., 2000*; *Jin et al., 2014*; *Kravitz et al., 2010*; *Markowitz et al., 2018*; *Mink, 2003*; *Tecuapetla et al., 2016*). It is well known that D1- and D2-SPNs are spatially intermixed in the striatum, and they both receive major excitatory inputs from the cerebral cortex (*Bolam et al., 2000*; *Pan et al., 2010*). A previous monosynaptic rabies tracing study has revealed that sensory and limbic cortical regions preferably send projections to D1-SPNs, compared to the motor cortical inputs biased toward D2-SPNs (*Wall et al., 2013*). However, this anatomical analysis was based on relative percentage of various inputs to a certain population, D1 or D2, and does not reflect the absolute number of cortical projections or the crosstalk of projections to both pathways. Furthermore, how the functional distinction between these two pathways is generated in the corticostriatal circuitry, and whether the striatal D1- and D2-SPNs receive the inputs from the same or different group of cortical neurons remains largely unknown. This is mainly due to the lack of appropriate tools to label and manipulate the specific cortical subpopulations projecting to D1- vs. D2-SPNs for functional investigations.

Here, using a G-deleted rabies system in mice (*Klug et al., 2018*; *Osakada et al., 2011*; *Wall et al., 2013*), we are able to selectively target and express channelrhodopsin-2 (ChR2) in presynaptic neurons projecting to D1- vs. D2-expressing SPNs. Whole-cell recordings from brain slice reveal that only one-third of the excitatory inputs to D1-SPNs target D2-SPNs, suggesting that many excitatory inputs to D1-SPNs selectively drive the direct pathway. In contrast, a large proportion of excitatory inputs to D2-SPNs send collateral projections to D1-SPNs, implying that excitatory inputs to D2-SPNs control both the indirect and direct pathways. Optogenetic stimulation of D1- vs. D2-SPN-projecting cortical neurons in vivo differently regulates locomotion, reinforcement learning, and sequence behavior, in a cell-type and brain-region-dependent manner. These results reveal the functional organization of cell-type- and pathway-specific corticostriatal subcircuits, providing essential insights into how they might control behavior in health and disease.

## Results

A modified rabies virus system (*Klug et al., 2018*; *Osakada et al., 2011*; *Wall et al., 2013*) was employed to label and functionally target the specific cortical neurons projecting to striatal D1- versus D2-SPNs. Specifically, D1- or A2a-Cre mice (*Gong et al., 2007*) were injected with Cre-dependent helper viruses (AAV5/EF1α-Flex-TVA-mCherry, AAV8/CA-Flex-RG) in the dorsal striatum (*Klug et al., 2018*; *Wall et al., 2013*; *Figure 1A–B*; see Materials and Methods). Three weeks later, either (EnvA) SAD-ΔG Rabies-GFP or (EnvA) SAD-ΔG Rabies-ChR2-mCherry was injected into the same striatal location to retrogradely infect the presynaptic cortical neurons projecting to D1- or D2-SPNs (*Figure 1B*). We first injected (EnvA) SAD-ΔG Rabies-GFP in a subgroup of mice to validate the corticostriatal anatomy. In both D1- and A2a-Cre tracing experiments, intensive labeling was found in different cortical regions as expected, including the midcingulate cortex (MCC; *van Heukelum et al., 2020*; *Vogt and Paxinos, 2014*) and the primary motor cortex (M1), which targets mainly the dorsal medial and dorsal lateral striatum, respectively (*Aoki et al., 2019*; *Bolam et al., 2000*; *Pan et al., 2010*; *Shepherd, 2013*; *Figure 1C and D*). For functional studies, (EnvA) SAD-ΔG Rabies-ChR2-mCherry was utilized to express ChR2 in the presynaptic cortical neurons projecting to D1- or D2-SPNs. To validate the functional expression of ChR2 in the cortex, whole-cell patch clamp recordings were performed from the mCherry-positive layer V pyramidal neurons in M1 around day 7 post rabies injection (*Figure 1E–G*; see Materials and methods). Both the current-voltage relationship revealed by somatic current injections (*Figure 1H*) and the spiking activity elicited by blue laser frequency stimulation (*Figure 1I*; *Figure 1—figure supplement 1*) confirmed the overall health and the functional expression of ChR2 in the rabies-infected cortical neurons. These results thus demonstrate that we were able to successfully target and functionally express ChR2 in presynaptic cortical neurons projecting to either striatal D1- or D2-SPNs.

Taking advantage of this rabies-ChR2 system, we first sought to determine how many functional excitatory inputs that the striatal D1- and D2-SPNs might share. The possible functional organization of excitatory inputs to D1- and D2-SPNs at the single cell level, like the corticostriatal projections,

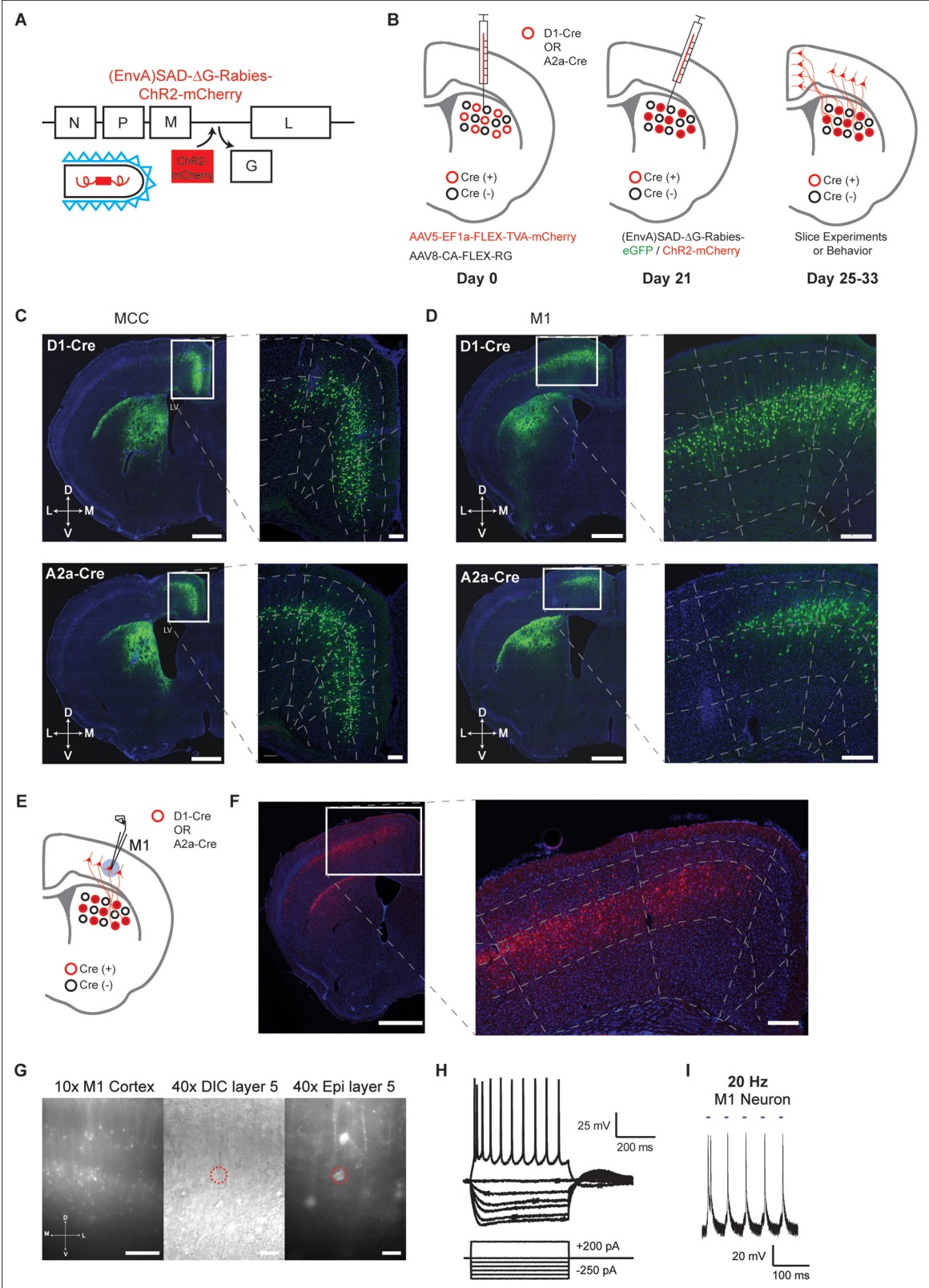

**Figure 1.** Selective labeling and functional expression of ChR2 in specific cortical neurons projecting to striatal D1- vs. D2-SPNs. (**A**) Schematic of SAD-ΔG-Rabies-ChR2-mCherry construct with the glycoprotein deleted and replaced with ChR2-mCherry. (**B**) Timeline of viral injections of Cre-dependent helper viruses and the modified rabies virus for slice and behavioral experiments. (**C**) Example of coronal brain section with rabies-eGFP injection in the dorsal medial striatum of D1-Cre (top) or A2a-Cre (bottom) mouse shows enriched eGFP expression in the MCC. Scale bar, 1 mm. Inset (right):

*Figure 1 continued on next page*

*Figure 1 continued*

Higher magnification of retrogradely labeled striatal D1- or D2-SPN projecting neurons in the MCC expressing eGFP. Dotted lines demarcate cortical lamina. Scale bar, 200 μm. (**D**) Similar experiments of labeling striatal D1- vs. D2-SPN projecting neurons in M1 with rabies-eGFP. (**E**) Cartoon brain schematic of ChR2-mCherry expressing M1 neurons projecting to SPNs (red) during whole-cell patch clamp recordings. (**F**) Example of coronal brain section with rabies-ChR2-mCherry injection in the dorsal lateral striatum of A2a-Cre mouse. Scale bar, 1 mm. Inset (right): Higher magnification of retrogradely labeled striatal D2-SPN projecting neurons in the M1 showed clear membrane expression of ChR2-mCherry. Scale bar, 200 μm. (**G**) (left) 10× epifluorescent (red channel) of ChR2-mCherry-positive neurons in M1. Scale bar, 250 μm. (middle) 40× image of a patched layer 5 pyramidal neuron under DIC optics. Scale bar, 50 μm. (right) Epifluorescent image (red channel) showing patched layer 5 pyramidal neuron somas expressing ChR2-mCherry signal. Red dotted line denotes patched neuron. Scale bar, 50 μm. (**H**) Current-voltage traces of a ChR2-mCherry positive layer 5 M1 neuron under current clamp responding to hyperpolarizing and depolarizing current injection steps. Scale bars, 200 ms, 25 mV. (**I**) Optogenetic stimulation (20 Hz) elicits robust action potentials with high fidelity in a ChR2-mCherry positive D1-SPN projecting M1 neuron in layer 5. Scale bars, 100 ms, 20 mV.

The online version of this article includes the following source data and figure supplement(s) for figure 1:

**Source data 1.** Current-clamp recording of a rabies-labeled cortical neuron expressing ChR2-mCherry.

**Figure supplement 1.** Optogenetic stimulation at 5 Hz reliably evoked action potential firing in a ChR2-expressing M1 neuron labeled by rabies infection.

could be completely segregated, totally overlapping, or partially mixed (*Figure 2A*). In order to differentiate these possibilities, we made whole-cell recordings from D1- or D2-SPNs in brain slice by optogenetic stimulation of rabies-ChR2-infected excitatory terminals in striatum. We asked what the probability is that a D1- or D2-SPN is targeted by the same presynaptic excitatory inputs projecting to the nearby D1- or D2-SPN population. D1- or A2a-Cre mice were crossed to the D1- or D2-eGFP reporter line for visualizing striatal D1- vs. D2-SPNs in slice recordings (see Materials and methods). Following the helper viruses and rabies-ChR2-mCherry injection in the D1-/A2a-Cre x D1-/D2-eGFP mice, the mCherry negative striatal SPNs were selected to be recorded in the whole-cell mode and D1- vs. D2-SPNs can be further separated based on the eGFP expression. Picrotoxin, a GABA$_A$ antagonist, was added throughout the recordings to isolate the excitatory postsynaptic currents (EPSCs). Following the blue laser stimulation of ChR2-positive presynaptic terminals in striatum, the short-latency (<10 ms) EPSCs recorded were considered as the direct excitatory inputs on D1- or D2-SPNs (*Klug et al., 2018*; *Kress et al., 2013*), which can be blocked by glutamate antagonists NBQX/APV (see Materials and methods).

Recordings from the mCherry-negative, non-starter striatal D1-SPNs in striatal D1-rabies-ChR2-infected mice revealed that with high probability (~63%) a D1-SPN receives the inputs from the presynaptic excitatory neurons projecting to surrounding D1-SPNs (*Figure 2B and D*; *Figure 2—figure supplement 1*). This is true from recordings in non-starter D1-SPNs identified both as mCherry (−) / eGFP (+) in D1-Cre x D1-eGFP mice and mCherry (−) / eGFP (−) in D1-Cre x D2-eGFP mice (*Figure 2B and D*). Similarly, recordings from mCherry-negative non-starter striatal D2-SPNs in striatal D2-rabies-ChR2-tracing mice revealed that with a very high probability (~79%) a D2-SPN receives the inputs from the presynaptic excitatory neurons that project to surrounding D2-SPNs (*Figure 2C and D*; *Figure 2—figure supplement 1*). Again, it is similar to recordings in non-starter D2-SPNs identified both as mCherry (−) / eGFP (+) in A2a-Cre x D2-eGFP mice and mCherry (−) / eGFP (−) in A2a-Cre x D1-eGFP mice (*Figure 2C and D*). However, recordings from striatal D2-SPNs in the striatal D1-rabies-ChR2-tracing mice revealed that the chance for a D2-SPN to receive excitatory inputs from the presynaptic neurons projecting to surrounding D1-SPNs is rather low (~40%, *Figure 2E and G*; *Figure 2—figure supplement 1*). In contrast, recordings from striatal D1-SPNs in the striatal D2-rabies-ChR2-tracing mice revealed that the chance for a D1-SPN to receive the excitatory inputs from the presynaptic neurons projecting to surrounding D2-SPNs is remarkably high (~73%, *Figure 2F and G*; *Figure 2—figure supplement 1*). These data unveil a complex picture including both parallel and crosstalk between the excitatory inputs to D1- and D2-SPNs. Notably, the likelihood that the input connectivity was significantly higher from the presynaptic excitatory inputs of D2-SPNs to D1-SPNs than from the presynaptic excitatory inputs of D1-SPNs to D2-SPNs (*Figure 2D and G*). Together, these results suggest largely segregated yet asymmetrically overlapping excitatory projections to striatum where the majority of excitatory inputs to D1-SPNs only target the D1-SPNs, while most excitatory inputs to D2-SPNs target both D2- and D1-SPNs.

Based on this asymmetrically overlapping functional organization, one would predict that the excitatory inputs to D1-SPNs mostly control the striatal direct pathway, while the inputs to D2-SPNs would

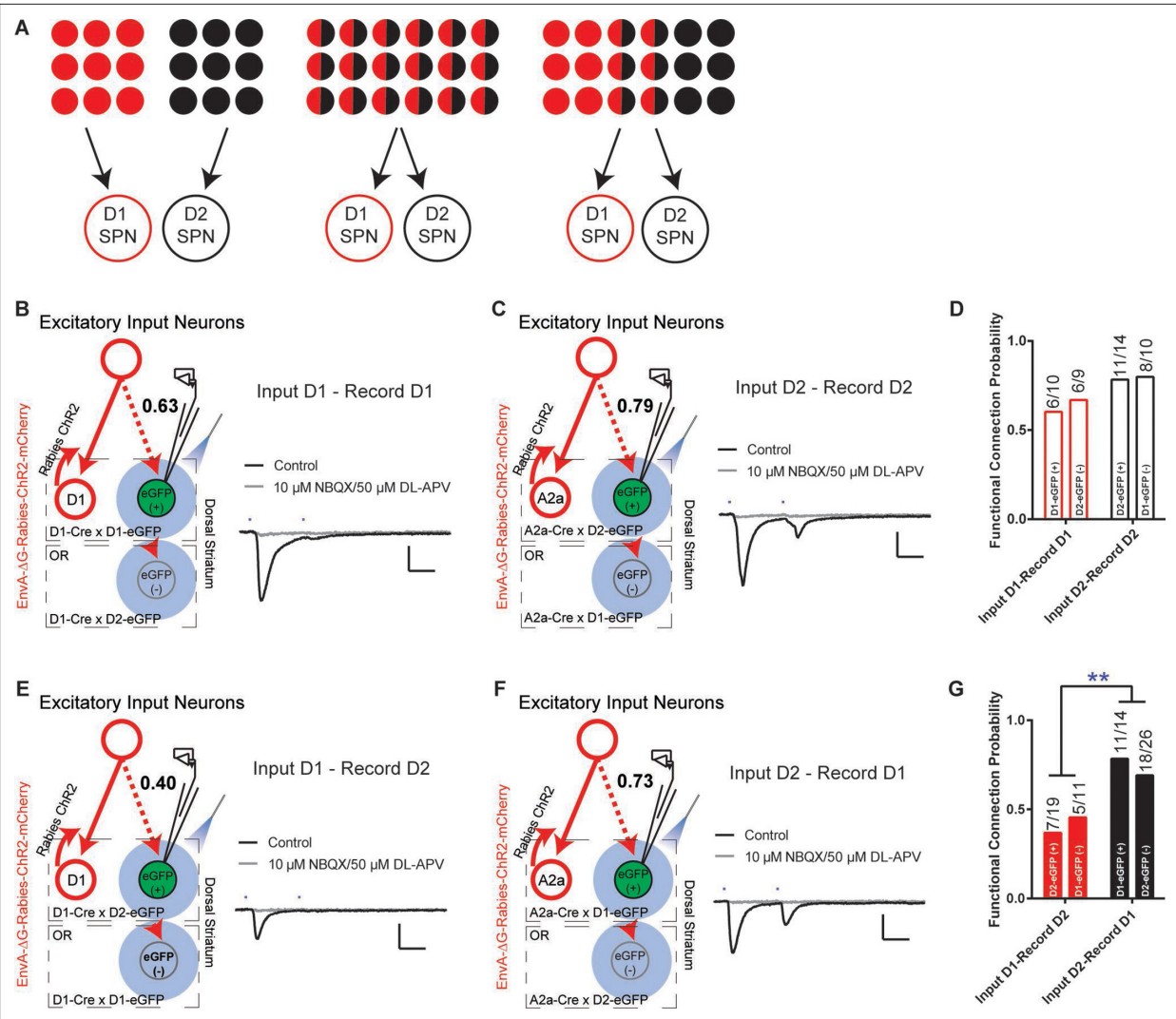

**Figure 2.** The excitatory inputs to striatal D1- vs. D2-SPNs are partially segregated with asymmetrical overlapping. (**A**) Schematic for the possible organization of the excitatory inputs to striatal D1- vs. D2-SPNs from completely segregated (left), totally overlapping (middle), to partially mixed (right). The red and black filled circles indicate the individual neurons projecting to D1- vs. D2-SPNs, respectively. The half red and half black circles imply the neurons projecting to both. (**B**) (left) Schematic of rabies-ChR2 labeling of the inputs to D1-SPNs and whole-cell recordings of rabies-negative striatal D1-SPNs with local optogenetic stimulation. (right) Example of the average EPSC trace showing short latency response to paired pulses (50 ms ISI) stimulation (black), that is blocked by AMPAR and NMDAR antagonists (gray). All recordings were conducted in the presence of picrotoxin (PTX) to isolate excitatory transmission. Scale bar, 25 ms, 100 pA. Same conditions applied to all following recordings. (**C**) Whole-cell recording of rabies-negative striatal D2-SPNs with local optogenetic stimulation with rabies-ChR2 labeling of the inputs to D2-SPNs. (**D**) The likelihood of the inputs to D1-SPNs forming a functional connection with nearby non-starter D1-SPNs, and the likelihood of the D2-SPN situation. Numbers above the bars denote the number of cells that show functional connectivity within total recorded. Fisher's exact test, p=0.3137. Number of animals in each group: D1-projecting to D1 EGFP(+), N=7; D1-projecting to D2 EGFP(−), N=8; D2-projecting to D2 EGFP(+), N=10; D2-projecting to D1 EGFP(−), N=8. (**E–F**) Whole-cell recording of rabies-negative striatal D2-SPNs with local optogenetic stimulation with rabies-ChR2 labeling of the inputs to D1-SPNs (**E**), and recording of rabies-negative D1-SPNs with stimulation of inputs to D2-SPNs (**F**). (**G**) The likelihood of the inputs to D1-SPNs forming a functional connection with nearby non-starter D2-SPNs, and the likelihood of the inputs to D2-SPNs forming a functional connection with nearby non-starter D1-SPNs. Fisher's exact test, p=0.0079. **, p<0.01. Number of animals in each group: D1-projecting to D2 EGFP(+), N=8; D1-projecting to D1 EGFP(−), N=7; D2-projecting to D1 EGFP(+), N=8; D2-projecting to D2 EGFP(−), N=10.

The online version of this article includes the following source data and figure supplement(s) for figure 2:

**Source data 1.** Number of connections from D1- or D2-projecting cortical neurons onto non-starter D1- or D2-SPNs.

**Figure supplement 1.** The synaptic properties of projections from D1- or D2-SPN retrogradely labeled cortical inputs to striatal D1- or D2-SPNs.

drive both the indirect and direct pathways (*Figure 3A*). To test whether this is the case, we injected rabies-ChR2-mCherry into the dorsal striatum of D1- or A2a-Cre mice as before, and implanted optic fibers bilaterally in either MCC or M1 (see Materials and methods). This allows us to selectively activate D1- or D2-SPN projecting neurons in MCC or M1 and determine the optogenetic effects on behavior. For comparison, we performed behavioral experiments by optogenetic stimulation of striatal D1- or D2-SPNs in dorsal medial (DMS) and dorsal lateral striatum (DLS), two areas that receive dense excitatory projections from MCC and M1, respectively (*Aoki et al., 2019*; *Shepherd, 2013*; see Materials and methods). Consistent with the previous observations (*Kravitz et al., 2010*), optogenetic stimulation (20 Hz) of D1-SPNs in the DMS or DLS facilitated locomotion (*Figure 3B, C, E and F*). Conversely, optogenetic stimulation (20 Hz) of D2-SPNs in DMS significantly suppressed locomotion (*Figure 3B and D*), which is less obvious in DLS (*Figure 3E and G*).

Notably, high-frequency (20 Hz) but not low-frequency (5 Hz) optogenetic stimulation of MCC neurons that project to D1-SPNs significantly facilitated locomotion in the open field (*Figure 3H, I*; *Figure 3—figure supplement 1*), similar to D1-SPN activation in DMS. However, optogenetic stimulation (20 Hz) of D2-SPN projecting MCC neurons in the same location did not alter locomotion in the open field (*Figure 3H and J*), in contrast with the effects of stimulation of D2-SPNs in DMS (*Figure 3D*). Similarly, high-frequency optogenetic stimulation (20 Hz) of M1 neurons that project to D1-SPNs facilitated locomotion in the open field (*Figure 3K and L*; *Figure 3—figure supplement 1*), while 20 Hz stimulation of the M1 neurons projecting to D2-SPNs did not significantly alter locomotion (*Figure 3K and M*). Further control experiments employing the same optogenetic stimulation in the exact cortical locations but with ChR2 expression only in the striatum do not generate any behavioral phenotypes (*Figure 3—figure supplement 2*). It thus rules out the possibility that the behavioral effects observed by cortical stimulation in the rabies-ChR2 mice were triggered through direct striatal activation due to the light penetration into the striatum. These results are consistent with the functional connectivity in which the excitatory inputs to D1-SPNs mostly drive the direct pathway, and the inputs to D2-SPNs target both the indirect and direct pathways (*Figure 3A*). It also suggests that the cortical neurons in the same cortical layer and spatial location could differently control actions depending on their striatal projection targets, in a pathway- and cell type-specific manner.

We next ask whether the cortical subpopulations projecting to striatal D1- vs. D2-SPNs could differently control action learning. We first performed experiments in the D1-Cre mice with viral expression of ChR2 in the striatum and found that optogenetic stimulation of D1-SPNs robustly supported intracranial self-stimulation (ICSS; *Figure 3N*) in either DMS (*Figure 3O*) or DLS (*Figure 3P*). Conversely, optogenetic stimulation of D2-SPNs, either in DMS (*Figure 3O*) or DLS (*Figure 3P*), did not promote ICSS behavior. These data confirmed that the D1-SPN activation in both DMS and DLS drives action learning and ICSS, while D2-SPN stimulation does not strongly support ICSS behavior (*Kravitz et al., 2012*; *Vicente et al., 2016*).

We then test how the striatum-projecting cortical neurons in MCC or M1 would support ICSS behavior, and whether there is any difference between activation of the D1- vs. D2-SPN projecting cortical neurons. Similar to the effects of direct striatal D1-SPN stimulation (*Figure 3O and P*), optogenetic stimulation of striatal D1-SPN projecting neurons was sufficient to support ICSS behavior both in MCC (*Figure 3Q and R*) and in M1 (*Figure 3Q and S*). Notably, optogenetic stimulation of the cortical neurons projecting to D2-SPNs also significantly drove ICSS behavior, irrespective of whether it is in MCC (*Figure 3R*) or M1 (*Figure 3S*). These data suggested that optogenetic activation of either D1- or D2-SPN projecting neurons in MCC or M1 could drive reinforcement learning and support ICSS behavior.

Corticostriatal circuitry is critical for action sequence learning and execution (*Geddes et al., 2018*; *Hikosaka et al., 1998*; *Jin and Costa, 2010*; *Jin and Costa, 2015*; *Jin et al., 2014*; *Tanji, 2001*; *Tecuapetla et al., 2016*). In particular, striatal direct and indirect pathways have been suggested to play distinct roles in controlling learned action sequences, as D1-SPNs facilitate ongoing actions while D2-SPNs inhibit actions and mediate switching (*Geddes et al., 2018*; *Jin and Costa, 2010*; *Jin and Costa, 2015*; *Jin et al., 2014*; *Tecuapetla et al., 2016*). We thus ask how the D1- vs. D2-SPN projecting neurons in MCC and M1 regulate the learned action sequences. D1- or A2a-Cre mice injected with helper viruses were trained under fixed-ratio schedule, in which a fixed amount of eight (FR8) leads to reward (*Geddes et al., 2018*; *Jin and Costa, 2010*; *Jin et al., 2014*; *Tecuapetla et al., 2016*; *Figure 4A*; see Materials and methods). Three weeks later, the trained animals were injected

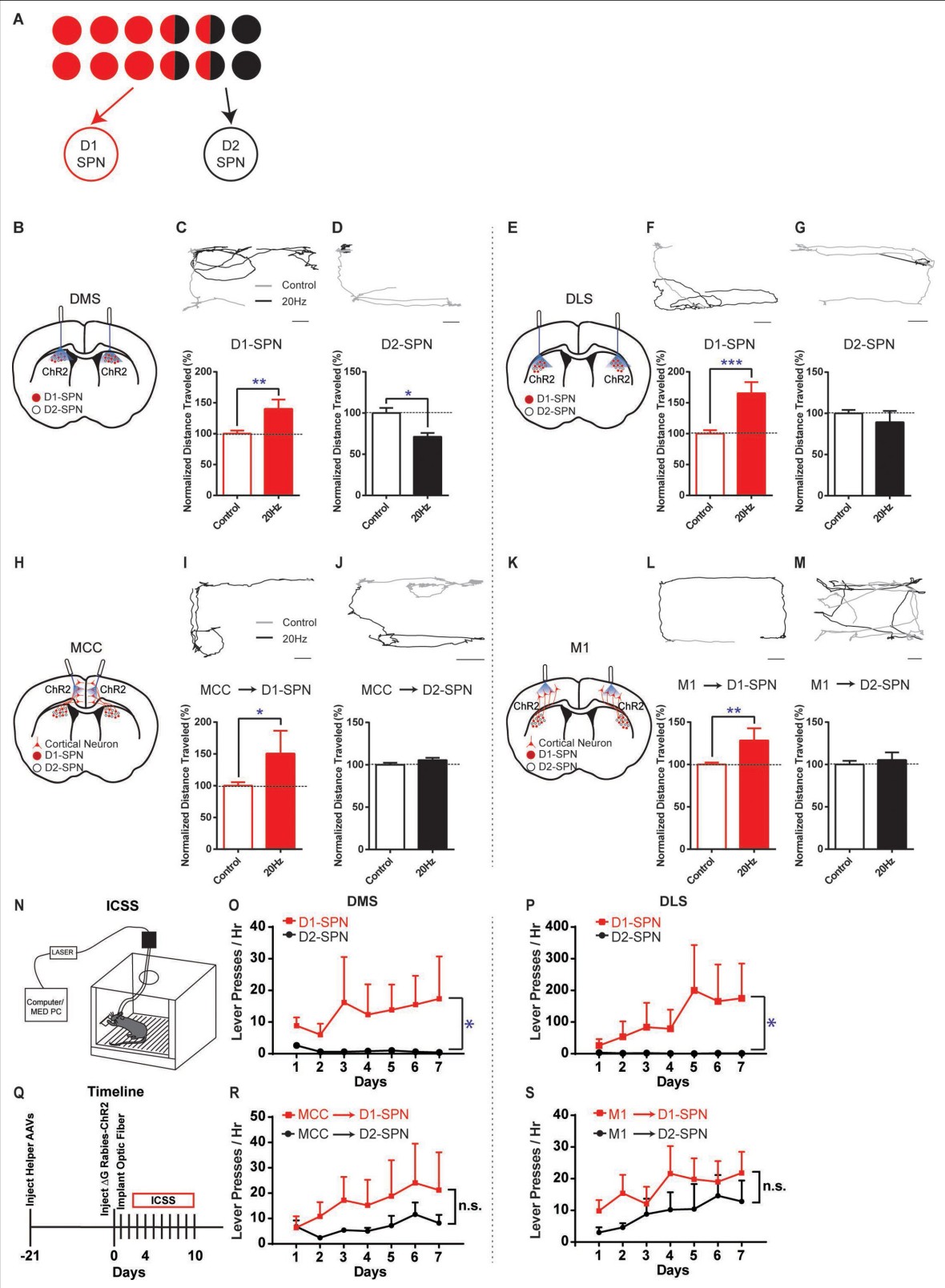

**Figure 3.** Different effects of optogenetic stimulation of D1- vs. D2-SPN projecting cortical neurons on locomotion and reinforcement learning. (**A**) Schematic of largely segregated yet partially overlapping excitatory inputs to striatal D1- vs. D2-SPNs. (**B**) Schematic of dorsal medial striatum (DMS) injection of Cre-dependent AAV-ChR2 and optogenetic stimulation in D1- or A2a-Cre mice. (**C**) (top) Example of locomotion path under control (black) and following 20 Hz optogenetic stimulation (gray) of DMS D1-SPNs in open field. Scale bar, 5 cm, same for below. (bottom) Stimulation of

*Figure 3 continued on next page*

*Figure 3 continued*

D1-SPNs in DMS facilitates locomotion (n=5, unpaired two-tailed *t*-test, *t*=3.386, p=0.0046). **, p<0.01. Data are expressed as mean ± SEM. (**D**) 20 Hz stimulation of D2-SPNs in DMS suppresses locomotion (n=5, unpaired two-tailed *t*-test, *t*=2.559, p=0.0227). *, p<0.05. Data are expressed as mean ± SEM. (**E**) Schematic for dorsal lateral striatum (DLS) optogenetics. (**F–G**) 20 Hz stimulation of D1-SPNs in DLS facilitates locomotion (**F**, n=5, unpaired two-tailed *t*-test, *t*=4.736, p=0.0003), while stimulation of D2-SPNs in DLS does not significantly suppress locomotion in open field (**G**, n=5, unpaired two-tailed *t*-test, *t*=1.026, p=0.3224). ***, p<0.001. Data are expressed as mean ± SEM. (**H**) Schematic of rabies-ChR2 labeling of the inputs to D1 or D2-SPNs and optogenetic stimulation in MCC. (**I–J**) 20 Hz stimulation of MCC neurons projecting to D1-SPNs facilitates locomotion (**I**, n=9, unpaired two-tailed *t*-test, *t*=2.344, p=0.0344), while stimulation of MCC neurons projecting to D2-SPNs does not alter locomotion (**J**, n=10, unpaired two-tailed *t*-test, *t*=1.214, p=0.2447). *, p<0.05. Data are expressed as mean ± SEM. (**K**) Schematic of rabies-ChR2 labeling of the inputs to D1 or D2-SPNs and optogenetic stimulation in M1. (**L–M**) 20 Hz stimulation of the M1 neurons projecting to D1-SPNs facilitates locomotion (**L**, n=7, unpaired two-tailed *t*-test, *t*=3.276, p=0.0055), while stimulation of the M1 neurons projecting to D2-SPNs does not significantly alter locomotion (**M**, n=8, Unpaired two-tailed *t*-test, *t*=0.5796, p=0.5714). **, p<0.01. Data are expressed as mean ± SEM. (**N**) Schematic of a mouse performing intracranial self-stimulation (ICSS) behavior. (**O–P**) D1-SPN (red) but not D2-SPN stimulation (black) drives ICSS behavior in both the DMS (**O**: D1, n=6, permutation test, slope = 1.5060, p=0.0378; D2, n=5, permutation test, slope = –0.2214, p=0.1021; one-tailed Mann Whitney test, Day 7 D1 vs. D2, p=0.0130) and the DLS (**P**: D1, n=6, permutation test, slope = 28.1429, p=0.0082; D2, n=5, permutation test, slope = –0.3429, p=0.0463; one-tailed Mann Whitney test, Day 7 D1 vs. D2, p=0.0390). *, p<0.05. Data are expressed as mean ± SEM. (**Q**) Timeline of helper virus injections, rabies-ChR2 injections and optogenetic stimulation for ICSS behavior. (**R–S**) Optogenetic stimulation of the cortical neurons projecting to either D1- or D2-SPNs induces ICSS behavior in both the MCC (**R**: MCC-D1, n=5, permutation test, Day1-Day7, slope = 2.5857, p=0.0034; MCC-D2, n=5, Day2-Day7, permutation test, slope = 1.4229, p=0.0344; no significant effect on Day7, MCC-D1 vs. MCC-D2, two-tailed Mann Whitney test, p=0.9999) and the M1 (**S**: M1-D1, n=5, permutation test, Day1-Day7, slope = 1.8214, p=0.0259; M1-D2, n=5, Day1-Day7, permutation test, slope = 1.8214, p=0.0025; no significant effect on Day7, M1-D1 vs. M1-D2, two-tailed Mann-Whitney test, p=0.3810). n.s., not statistically significant. Data are expressed as mean ± SEM.

The online version of this article includes the following source data and figure supplement(s) for figure 3:

**Source data 1.** Effect of 20 Hz optogenetic stimulation on normalized distance traveled for D1- or D2-SPNs in DMS and DLS.

**Source data 2.** Normalized distance traveled during 20 Hz optogenetic stimulation of MCC neurons projecting to D1- or D2-SPNs.

**Source data 3.** Normalized distance traveled during 20 Hz optogenetic stimulation of M1 neurons projecting to D1- or D2-SPNs.

**Source data 4.** ICSS behavior induced by 20 Hz optogenetic stimulation of D1- or D2-SPNs in the DMS and DLS.

**Source data 5.** ICSS behavior induced by 20 Hz optogenetic stimulation of D1- or D2-projecting neurons in the MCC and M1.

**Figure supplement 1.** Low-frequency (5 Hz) optogenetic stimulation of cortical neurons projecting to striatal D1- or D2-SPNs has little effect on locomotion activity.

**Figure supplement 1—source data 1.** Normalized distance traveled during 5 Hz optogenetic stimulation of MCC neurons projecting to D1- or D2-SPNs.

**Figure supplement 1—source data 2.** Normalized distance traveled during 5 Hz optogenetic stimulation of M1 neurons projecting to D1- or D2-SPNs.

**Figure supplement 2.** No effects of optogenetic stimulation of M1 on locomotion in mice with ChR2 expression in either D1- or D2-SPNs of DMS.

**Figure supplement 2—source data 1.** Normalized distance traveled during 20 Hz optogenetic stimulation of M1 with ChR2 expressed exclusively in striatal D1- or D2-SPNs.

with (EnvA) SAD-ΔG Rabies-ChR2-mCherry virus in the dorsal striatum and optic fibers were bilaterally implanted in either MCC or M1 as before. Mice were continuously trained for around 4 days after 4 days of recovery from the surgeries to allow the rabies-mediated ChR2 expression before the optogenetic experiments start (*Figure 4E*). High-frequency stimulation (20 Hz) of the cortical neurons projecting to D1-SPNs or D2-SPNs was delivered upon the first lever press of the FR8 sequence in randomly chosen 50% trials (*Geddes et al., 2018*; *Tecuapetla et al., 2016*; *Figure 4A and E*, see Materials and methods). Stimulation of MCC inputs to D1-SPNs facilitated lever pressing over the duration of the FR8 sequence (*Figure 4B and D*). Conversely, stimulation of MCC inputs to D2-SPNs slightly reduced the lever press rate over the stimulation period (*Figure 4C and D*). The modulation effects on lever pressing rate were significantly different between optogenetic stimulation of D1- and D2-SPN projecting MCC neurons (*Figure 4D*). On the other hand, optogenetic activation of the M1 neurons that project to D1-SPNs facilitated lever pressing during sequence execution (*Figure 4F and H*), similar to the effects of MCC stimulation. However, optogenetic stimulation of the M1 neurons projecting to D2-SPNs delivered an overall facilitation effect on lever pressing (*Figure 4G and H*). Overall, stimulation of either D1- or D2-SPN projecting M1 neurons facilitated lever pressing to a similar degree (*Figure 4H*). These results thus revealed the highly heterogeneous functions of corticostriatal subcircuits in controlling learned action sequences, depending on both the cortical region and their cell-type-specific targets in striatum.

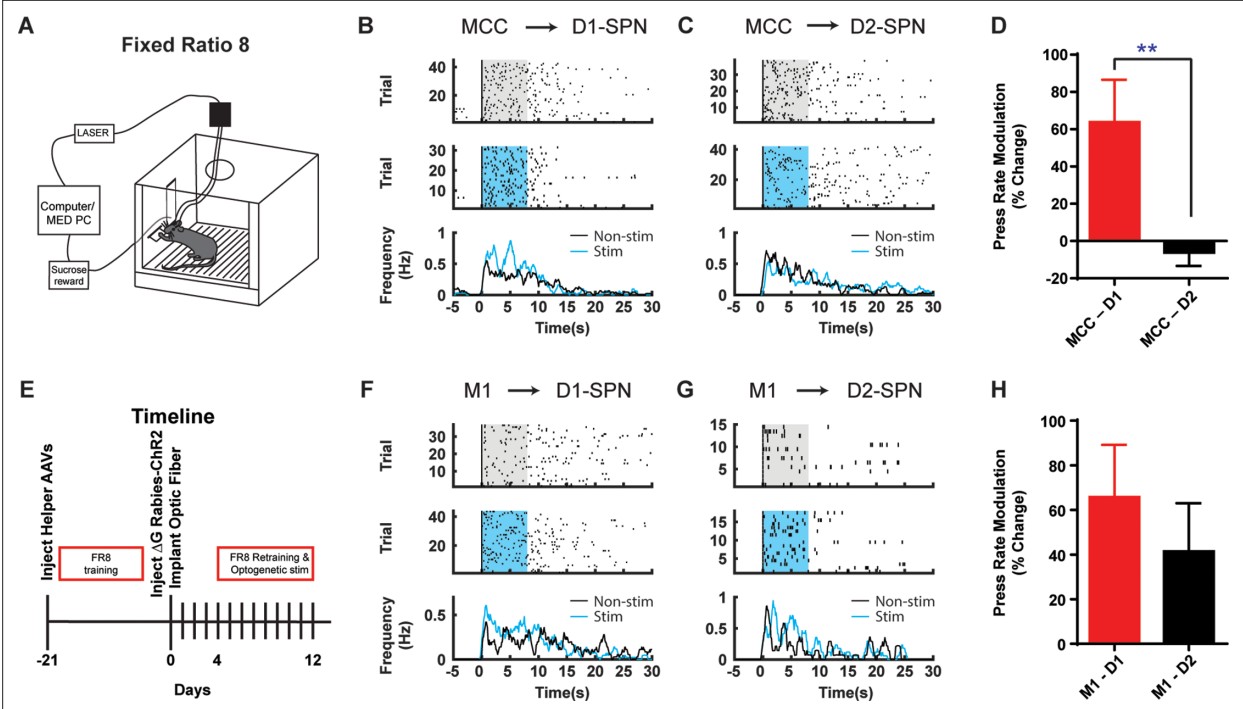

**Figure 4.** Optogenetic stimulation of D1- vs. D2-SPN projecting cortical neurons differently modulates action sequence execution. (**A**) Schematic of a mouse performing FR8 sequence. (**B**) Optogenetic stimulation (20 Hz) of the D1-SPN projecting MCC neurons during FR8 sequence. Example lever pressing (black bar) in control (top) vs. stimulation (middle) trials aligned to the first press, where the blue transparent rectangle corresponds to the window of optogenetic stimulation (20 Hz, 8 s). The black and blue lines in the PETH (bottom) indicate the lever press rate for control and stimulation conditions, respectively, same for below. (**C**) Optogenetic stimulation (20 Hz) of the D2-SPN projecting MCC neurons during FR8 sequence. (**D**) Average percent lever press rate change during optogenetic stimulation of D1- vs. D2-SPN projecting MCC neurons compared to control (MCC – D1, n=8; MCC – D2, n=7; Unpaired two-tailed *t*-test, *t*=2.774, p=0.0097). **, p<0.01. Effects compared to a theoretical percentage change of 0 of each individual manipulation (MCC-D1, n=8, one-sample two-tailed *t*-test, *t*=2.814, p=0.0131, 95% CI, 15.49–112.2; MCC-D2, n=7, one-sample two-tailed *t*-test, *t*=0.8481, p=0.4117, 95% CI, –21.78–9.502). Data are expressed as mean ± SEM. (**E**) Timeline of helper virus injections, rabies-ChR2 injections and optogenetic stimulation during action sequence performance. (**F–G**) Optogenetic stimulation (20 Hz) of the D1- (**F**) or D2-SPN (**G**) projecting M1 neurons during FR8 sequence. (**H**) Average percent lever press rate change during optogenetic stimulation of D1- vs. D2-SPN projecting M1 neurons compared to control (M1 – D1, n=6; M1 – D2, n=7; Unpaired two-tailed *t*-test, *t*=0.7651, p=0.4511). Effects compared to a theoretical percentage change of 0 of each individual manipulation (M1-D1, n=6, one-sample two-tailed Wilcoxon signed-rank test, p=0.0046, 97.75% CI, –0.7866–151.0; M1-D2, n=7, one-sample two-tailed Wilcoxon signed-rank test, p=0.0479, 96.48% CI, 2.350–62.86). Data are expressed as mean ± SEM.

The online version of this article includes the following source data for figure 4:

**Source data 1.** Change in sequence pressing during 20 Hz optogenetic stimulation of MCC or M1 neurons projecting to striatal D1- or D2-SPNs.

## Discussion

By taking advantage of a monosynaptic rabies tracing with optogenetics system, we have discovered a significant degree of segregation between the excitatory inputs to striatal D1- vs. D2-SPNs. Notably, the results unveiled an overall asymmetric crosstalk from the excitatory inputs of D2-SPNs onto D1-SPNs, but not vice versa. Striatal D1- and D2-SPNs receive excitatory inputs from both the cortex and thalamus (**Klug et al., 2018**; **Wall et al., 2013**). Since the current techniques do not allow us to isolate the inputs from a specific region to D1- vs. D2-SPNs in slice recording, these results do not exclude the possibility that there might be certain cortical or thalamic regions targeting D1- and D2-SPNs equally or even with a reverse bias. However, the overall functional organization does imply that while the excitatory inputs to D1-SPNs in general drive the striatal direct pathway, the excitatory inputs to D2-SPNs control both the striatal direct and indirect pathways. Indeed, it has been recently reported that corticospinal neurons, which project to both spinal cord and DLS, form uneven synapses onto direct and indirect pathway neurons in the DLS and preferentially target D1-other than D2-SPNs (**Nelson et al., 2021**). Furthermore, a series of in vivo optogenetic experiments in both MCC and M1 have further supported this notion and demonstrated that the functionally

heterogeneous corticostriatal neuronal subpopulations differently control actions, in both a cortical-region- and striatal-targeting-cell-type-specific manner. These in vivo functional findings in corticostriatal pathways are consistent with the observations of in vitro synapse connection probability. Future studies should aim to further dissect the organization and function of pathway-specific thalamostriatal subcircuits and determine whether they share the same principles of corticostriatal projections.

Our findings show that stimulation of D1-projecting cortical neurons produced behavioral effects closely resembling those of selective D1 activation in both open field and ICSS tests. This is consistent with our slice recording data, which revealed that D1-projecting cortical neurons exhibit a higher connection probability with D1-SPNs than with D2-SPNs. In contrast, interpreting the effects of D2-projecting cortical neuron stimulation is inherently more nuanced. In the open field test, activation of these neurons did not significantly modulate local motion. This could reflect a balanced influence of D1 activation, which facilitates movement, and D2 activation, which suppresses it - resulting in a net neutral behavioral outcome. In the ICSS test, the absence of a strong reinforcement effect typically associated with D2 activation, combined with partial reinforcement likely due to concurrent D1 activation, suggests that stimulation of D2-projecting neurons produces a mixed behavioral signal. This outcome supports the interpretation that these neurons synapse onto both D1- and D2-SPNs, leading to a blended behavioral response that differs from selective D1 or D2 activation alone. Together, these two behavioral assays offer complementary perspectives, providing a more complete view of how projection-specific cortical inputs influence striatal output and behavior.

In *Figure 4* of the current manuscript, we show that optogenetic activation of MCC neurons projecting to D1-SPNs facilitates sequence lever pressing, whereas activation of MCC neurons projecting to D2-SPNs does not induce significant behavioral changes. Conversely, activation of M1 neurons projecting to either D1- or D2-SPNs enhances lever pressing sequences. These observations align with our prior findings (*Geddes et al., 2018*; *Jin et al., 2014*), where we demonstrated that in the striatum, D1-SPN activation facilitates ongoing lever pressing, whereas D2-SPN activation is more involved in suppressing ongoing actions and promoting transitions between sub-sequences. Taken together, the facilitation of lever pressing by D1-projecting MCC and M1 neurons is consistent with their preferential connectivity to D1-SPNs and their established behavioral role.

What is particularly intriguing, though admittedly more complex, is the behavioral divergence observed upon activation of D2-SPN-projecting cortical neurons. Activation of D2-projecting MCC neurons does not alter lever pressing, possibly reflecting a counterbalancing effect from concurrent D1- and D2-SPN activation. In contrast, stimulation of D2-projecting M1 neurons facilitates lever pressing, albeit less robustly than their D1-projecting counterparts. This discrepancy may reflect regional differences in striatal targets, DMS for MCC versus DLS for M1, as also supported by our open field test results. Furthermore, our recent findings (*Zhang et al., 2025*) show that synaptic strength from Cg to D2-SPNs is stronger than to D1-SPNs, whereas the M1 pathway exhibits the opposite pattern. These data suggest that beyond projection ratios, synaptic strength also shapes cortico-striatal functional output. Thus, stronger D2-SPN synapses in the DMS may offset D1-SPN activation during MCC-D2 stimulation, dampening lever pressing increase. Conversely, weaker D2 synapses in the DLS may permit M1-D2 projections to facilitate behavior more readily.

To identify non-starter D1- or D2-SPNs, we used mCherry to label both TVA helper virus and ChR2-expressing rabies constructs, allowing us to record mCherry-negative non-starter SPNs labeled by GFP to calculate connection probability. However, this approach introduces a limitation: it is not possible to accurately assess rabies infection efficiency, because both TVA-only SPNs and monosynaptically traced SPNs exhibit mCherry signal in addition to true starter SPNs. Existing data indicate that the connection probabilities from D1-projecting cortical neurons to both D1- and D2-SPNs are lower than those from D2-projecting cortical neurons to both SPN subtypes. One possible explanation is that rabies virus replicates more efficiently in D1-SPNs than in D2-SPNs, or vice versa, causing the calculated connection ratios to reflect a combination of true connectivity and differences in viral replication efficiency. On the other hand, we recorded from non-starter SPNs within the injection area that were surrounded by mCherry-positive starter cells, which should share similar topographic cortical projections. Under these conditions, the number of labeled starter SPNs is unlikely to substantially influence the connection probability measured in brain slices. Furthermore, our recent study (*Li and Jin, 2023*) proposes a 'triple-control' model of striatal function, consisting of a direct pathway operating linearly and two indirect pathway layers serving modulatory roles. This framework suggests a

functional requirement for D2-SPNs in the indirect pathway to receive more coordinated cortical excitatory input than D1-SPNs, which is consistent with the connection data obtained in the present study.

Although monosynaptic rabies tracing has been widely applied in anatomical studies, its use in functional and in vivo experiments remains limited by toxicity. In our in vivo optogenetic behavioral experiments, we confined all stimulations to within 12 days post-infection, a period during which rabies-infected cells are reported to remain relatively healthy (*Osakada et al., 2011*). Nevertheless, toxicity may still compromise neuronal excitability, disrupt synaptic transmission, or even cause the loss of starter SPNs. As a result, the observed optogenetic effects could partly arise from biased circuit function due to such cellular alterations. Addressing this issue will require improved monosynaptic retrograde tracing tools with reduced toxicity and more reliable labeling efficiency.

The cortical neurons projecting to striatum mainly consist of layer 2/3 and layer 5 pyramidal cells (*Klug et al., 2018*; *Wall et al., 2013*), including both the intratelencephalic (IT) and pyramidal tract (PT) types of neurons (*Shepherd, 2013*). While some anatomical preference might exist (*Lei et al., 2004*), it has been found that both the striatal direct and indirect pathways receive functional inputs from both the IT and PT neurons (*Ballion et al., 2008*; *Kress et al., 2013*). Our rabies-ChR2 tracing system allows us to further separate the cortical inputs to striatal D1- vs. D2-SPNs and selectively stimulate these specific cortical subpopulations during behavior and learning. These results have further revealed the diversity of corticostriatal cell subtypes and underscored their heterogeneous functions in behavior. Although the behavioral phenotypes of optogenetic stimulation of different cortical neuronal subpopulations are largely consistent with their functional connectivity with the striatal D1- vs. D2-SPNs do not necessarily suggest the observed effects were mediated completely by striatum but not through their collaterals targeting other brain regions or spinal cord (*Nelson et al., 2021*; *Shepherd, 2013*). In addition, it has been known that both striatal direct and indirect pathways receive inhibitory inputs from certain GABAergic interneurons in motor cortices (*Melzer et al., 2017*). In our behavioral experiments with optogenetic stimulation in the motor cortex, there might be a possible contribution from these striatum-projecting cortical inhibitory neurons. However, given the nature of sparse distribution of the GABAergic interneurons in the cortex, it is unlikely that they dominate the observed behavioral phenotype (*Melzer et al., 2017*). Due to the limitations of current viral approaches, one might question whether our in vivo optogenetic experiments could inadvertently activate terminals from rabies-infected neurons in cortical or thalamic regions beyond the targeted area. However, a previous optogenetic study on thalamo-cortical synapses (*Cruikshank et al., 2010*) demonstrated that sustained optogenetic stimulation (500 ms) of ChR2-expressing presynaptic terminals induces transient, rather than sustained, action potential firing on the postsynaptic neurons. Considering the much longer stimulation durations used in our optogenetic experiments (15 s for open-field, 1 s for ICSS, and 8 s for the FR8/FR4), it provides compelling evidence that the behavioral effects of optogenetic stimulation are most likely driven by the excitation of ChR2-expressing D1- or D2-SPN projecting neurons in the targeted cortical area, rather than by efferent terminals from neurons located outside this region. Nevertheless, from the striatum point of view, the distinct behavior effect does strongly suggest that the specific information the direct vs. indirect pathway received from the cortex is somehow channeled, but at the same time, effectively coordinated by the cortex.

These results have important implications for how the corticostriatal circuitry controls actions in health and disease. The traditional model of the basal ganglia suggests that the direct and indirect pathways play opposing roles in facilitating and inhibiting action, respectively (*Albin et al., 1989*; *DeLong, 1990*; *Kravitz et al., 2010*), although one study (*Cui et al., 2021*) reported an alternative pattern, which may stem from differences in stimulation coordinates and fiber-optic diameter. More recent models of basal ganglia, however, propose that the direct pathway co-activates and cooperates with the indirect pathway, with the former activating the selected action and the latter inhibiting the competing actions (*Cui et al., 2013*; *Hikosaka et al., 2000*; *Jin et al., 2014*; *Mink, 1996*; *Tecuapetla et al., 2016*). Under more complicated behavior context, it has been previously reported that the striatal D1- and D2-SPNs are co-activated during the initiation of an action sequence but become largely segregated during the sequence performance (*Geddes et al., 2018*; *Jin et al., 2014*). More specifically, the various subpopulations of striatal D1- and D2-SPNs differently change their firing activity to support the start/stop of the sequence, the execution of the elemental actions, and the switch between subsequences (*Geddes et al., 2018*). These previous findings thus suggested that

the striatal direct and indirect pathways have to dynamically coordinate their activity throughout the performance of sequential actions (*Geddes et al., 2018*; *Hikosaka et al., 2019*; *Jin and Costa, 2015*; *Markowitz et al., 2018*; *Tecuapetla et al., 2016*).

But how are the dynamically different activities in the striatal direct and indirect pathways generated in the circuitry? Both the striatal direct and indirect pathways are driven by the excitatory inputs from the cerebral cortex and thalamus (*Bolam et al., 2000Gerfen et al., 2016*; *Pan et al., 2010*; *Wall et al., 2013*). However, whether or not they receive the projections from the same presynaptic neurons, and how the input information is channeled into the two pathways for proper action control, remains mostly unknown. The current study has revealed the largely segregated but asymmetrically overlapping organization of the cortical projections to striatal direct vs. indirect pathway. This specific corticostriatal organization provides a structural foundation for the striatal direct and indirect pathways to implement such a dynamic coordination of activity during sequence behavior (*Geddes et al., 2018*; *Hikosaka et al., 2019*; *Hikosaka et al., 1998*; *Jin and Costa, 2010*; *Jin and Costa, 2015*; *Jin et al., 2014*; *Markowitz et al., 2018*; *Tanji, 2001*; *Tecuapetla et al., 2016*). For instance, the dedicated cortical projections to striatal direct vs. indirect pathway are well suited for controlling sequence initiation and termination, where the activation of D1- and D2-SPNs is critical (*DeLong, 1990*; *Geddes et al., 2018*). On the other hand, the overlapping cortical projections to both striatal direct and indirect pathways could be crucial for action switching, which requires proper coordination of the two pathways to inhibit current action and activate the upcoming one (*DeLong, 1990*; *Geddes et al., 2018*). Our findings also predict that the striatal D1- vs. D2-SPN projecting neurons in the cerebral cortex would fire differently but activate in relation with each other during behavior. Future work should aim to understand how these cortical subpopulations behave and coordinate to control the striatal direct and indirect pathways for action learning and selection in health and disease (*Dalley and Robbins, 2017*; *Geddes et al., 2018*; *Hikosaka et al., 2000*; *Jin et al., 2014*; *Mink, 2003*; *Monteiro and Feng, 2017*; *Redgrave et al., 2010*; *Shepherd, 2013*).

# Materials and methods

**Key resources table**

| Reagent type (species) or resource | Designation | Source or reference | Identifiers | Additional information |
|---|---|---|---|---|
| Strain, strain background (*Mus musculus*) | Drd1-Cre | The Jackson Laboratory | stock # 030329; RRID:IMSR_JAX:030329 | maintained on a C57BL/6 J background |
| Strain, strain background (*Mus musculus*) | Adora2a-Cre | The Jackson Laboratory | stock # 036158; RRID:MMRRC_036158-UCD | maintained on a C57BL/6 J background |
| Strain, strain background (*Mus musculus*) | C57BL/6 J | The Jackson Laboratory | stock # 000664; RRID:IMSR_JAX:000664 | |
| Strain, strain background (*Mus musculus*) | D1-eGFP | MMRRC | MMRRC_000297-MU; RRID:MMRRC_000297-MU | GENSAT: X60 |
| Strain, strain background (*Mus musculus*) | D2-eGFP | MMRRC | MMRRC_00230-UNC; RRID:MMRRC_000230-UNC | GENSAT: S118 |
| Strain, strain background (*Adeno-associated virus*) | AAV5/EF1α-Flex-TVA-mCherry | UNC Viral Vector Core | RRID:SCR-002448 | $3$–$4.3 \times 10^{12}$ particles/mL |
| Strain, strain background (*Adeno-associated virus*) | AAV8/CA-Flex-RG | UNC Viral Vector Core | RRID:SCR-002448 | $1.2$–$4.3 \times 10^{12}$ particles/mL |
| Strain, strain background (*Pseudotyped rabies virus*) | (EnvA) SAD-ΔG Rabies-eGFP | Salk Vector Core | RRID:SCR_014847 | |
| Strain, strain background (*Pseudotyped rabies virus*) | (EnvA) SAD-ΔG Rabies-ChR2-mCherry | Salk Vector Core | RRID:SCR_014847 | |
| Strain, strain background (*Adeno-associated virus*) | AAV5-EF1α-DIO-ChR2(H134R)-mCherry | Salk Vector Core | RRID:SCR_014847 | |
| Chemical compound, drug | NBQX disodium salt hydrate | MilliporeSigma | Cat.#. N183 | 10 µM (final) |

*Continued on next page*

*Continued*

| Reagent type (species) or resource | Designation | Source or reference | Identifiers | Additional information |
|---|---|---|---|---|
| Chemical compound, drug | DL-APV | MilliporeSigma | Cat.#. A5282 | 50 µM (final) |
| Chemical compound, drug | Picrotoxin | MilliporeSigma | Cat.#. P1675 | 50–100 µM (final) |
| Chemical compound, drug | QX-314 | MilliporeSigma | Cat.#. L5783 | |
| Software, algorithm | MATLAB | MathWorks | RRID:SCR_001622 | |
| Software, algorithm | Prism | GraphPad | RRID:SCR_002798 | |
| Software, algorithm | Fiji / ImageJ | NIH | RRID:SCR_002285 | |
| Software, algorithm | pClamp 9.2 | Molecular Devices | RRID:SCR_011323 | |
| Software, algorithm | Illustrator | Adobe | RRID:SCR_010279 | |
| Software, algorithm | MED PC | MED Associates | RRID:SCR_012156 | |
| Other | Allen Reference Atlas | | RRID:SCR_013286 | |

## Animals

All procedures were approved by the Salk Institute Institutional Animal Care and Use Committee (IACUC; protocol 12–00032) and followed NIH guidelines for the care and use of laboratory animals. Group-housed male and female mice (2–6 months old) were used in this study. Animals were housed on a 12 hr dark/12 hr light cycle (dark from 6 pm to 6 am). Heterozygous Drd1-Cre (The Jackson Laboratory, stock # 030329, GENSAT: EY217) and Adora2a-Cre (The Jackson Laboratory, stock # 036158, GENSAT: KG139) mice were obtained from MMRRC and were backcrossed to C57BL/6 J mice, stock # 000664 (>9 generations; *Cui et al., 2013*; *Jin et al., 2014*; *Madisen et al., 2012*; *Tecuapetla et al., 2016*). BAC reporter lines D1-eGFP (MMRRC: MMRRC_000297-MU; GENSAT: X60) and D2-eGFP (MMRRC: MMRRC_00230-UNC; GENSAT: S118; *Gong et al., 2007*) were crossed to Drd1-Cre (D1-Cre) and Adora2a-Cre (A2a-Cre) mice to identify D1- and D2-SPNs for electrophysiological recordings.

## Surgery and viral injection

For G-deleted rabies-mediated retrograde tracing and functional determination (slice recordings; *Smith et al., 2016*), all surgeries were performed under aseptic conditions with animals anesthetized with ketamine (100 mg/kg) / xylazine (10 mg/kg) while mounted on a stereotaxic device (Kopf Instruments; Tujunga, CA). The skull was leveled at bregma and lambda, and a small hole was drilled at the coordinate (from bregma and midline) of AP +0.5 mm, ML ±1.8 mm. A Hamilton syringe (33-gauge needle) containing 1 µl freshly mixed AAV5/EF1α-Flex-TVA-mCherry (UNC Vector Core; Chapel Hill, NC) and AAV8/CA-Flex-RG (UNC Vector Core; Chapel Hill, NC) was slowly lowered to DV - 2.2 mm from the dura to target dorsal central striatum. The virus cocktail was injected slowly over ~10 min, and the needle was left in place for ~5 additional minutes afterwards. Then, the needle was slowly retracted over 5 min to reduce the virus from moving into the needle track. After injection, mice were sutured and returned to their home cage with ibuprofen (50 mg/kg/day) in their drinking water for the following four days. They were given 3 weeks to allow for maximal expression of helper viruses before they were injected with 1.5 µl of (EnvA) SAD-ΔG Rabies-eGFP or 1.5 µl of (EnvA) SAD-ΔG Rabies-ChR2-mCherry (Salk Vector Core, La Jolla, CA) on an angle (18°) to avoid labeling any neurons in the initial injection tract in the same target region. Injecting locations were identical in D1-Cre and A2a-Cre animals. All the injections were done unilaterally for anatomical and slice physiology experiments and bilaterally for behavioral experiments.

To prepare animals for optogenetic behavior experiments testing D1- or D2-SPN projecting cortical neurons, animals were anesthetized with isoflurane (4% induction, 1–2% maintenance) and locally injected with bupivacaine to numb the incision site. The animals received bilateral injections of helper virus (TVA, RG) as before in dorsal striatum. After ~21 days of pre-training and full body weight recovered (see Operant Conditioning), the skull was exposed again and cleaned with 4% $H_2O_2$ and UV-light etched with Opti-Bond All-in-One (Kerr, Orange, CA). Then 1.5 µl (EnvA) SAD-ΔG Rabies-ChR2-mCherry was bilaterally injected in each hemisphere using the same coordinates as before.

Then, custom-made, polished optical fibers (200 µm diameter, 0.37 NA; Thor Labs, Newton, NJ) were implanted in input regions: MCC (AP +0.2 mm, ML ±0.8 mm for skull holes, fibers penetrate into brain at 17° angle off midline with traveling distance of 1.3 mm, actual fiber tips target brain at AP +0.2 mm, ML ±0.4 mm, DV –1.2 mm) or M1 (AP +0.5 mm, ML ±1.2 mm, DV –0.5 mm). The fibers were secured with a light-curing composite (Tetric EvoFlow, Ivoclar Vivadent; Mississauga, ON). Finally, a layer of black dental cement (Lang Dental, Wheeling, IL) was added on the top of the previous cement to support and block laser light diffusion during stimulation. Animals were given ibuprofen in their drinking water for pain management during post-surgery recovery (4 days).

For striatal opto-ICSS and open field experiments, D1- or A2a-Cre mice were injected bilaterally with AAV5-EF1α-DIO-ChR2(H134R)-mCherry (Salk Vector Core, La Jolla, CA) in DMS (AP 0.5 mm, ML ±1.5 mm, DV –2.2 mm) or DLS (AP 0.5 mm, ML ±2.5 mm, DV –2.2 mm), and fiber optics were implanted ~0.2 mm above the injection site. In control experiments for testing striatal activation by light penetration from cortical optic fibers (*Figure 3—figure supplement 2*), D1- or A2a-Cre mice were injected with AAV5-EF1α-DIO-ChR2(H134R)-mCherry bilaterally in DMS, and fiber optics were bilaterally implanted into M1 of the same coordinates as previously described.

## Ex vivo brain slice electrophysiology

4–8 days were allowed for expression and optimal cell health post unilateral (EnvA) SAD-ΔG Rabies-ChR2-mCherry injection before electrophysiology recordings on acute slice were carried out (*Klug et al., 2018*; *Smith et al., 2016*). Mice were anesthetized with ketamine/xylazine and transcardially perfused with ~20 mL ice-cold, bubbling (95% $O_2$/5% $CO_2$) NMDG cutting solution [consisting of (in mM): NMDG 105, HCl 105, KCl 2.5, $NaH_2PO_4$ 1.2, $NaHCO_3$ 26, Glucose 25, Sodium L-Ascorbate 5, Sodium Pyruvate 3, Thiourea 2, $MgSO_4$ 10, $CaCl_2$ 0.5, 300 mOsm, pH = 7.4]. The extracted brain was blocked coronally with a brain matrix (Zivic Instruments; Pittsburg, PA) and acute coronal slices (300 µm) were cut on a vibratome (VT1000S, Leica Microsystems; Buffalo Grove, IL) through the striatum in ice-cold, bubbling NMDG-based cutting solution. Slices recovered for 15 min at 32 °C in bubbling NMDG cutting solution, then transferred to a holding chamber containing normal aCSF [consisting of (in mM): NaCl 125, KCl 2.5, $NaH_2PO_4$ 1.25, $NaHCO_3$ 25, D-Glucose 12.5, $MgCl_2$ 1, $CaCl_2$ 2, pH = 7.4, 295 mOsm] bubbling (95% $O_2$/5% $CO_2$) at 28 °C. At least 1 hr after recovery, the slices were placed in the recording chamber, in which normal aCSF (33~34 °C, bubbling with 95% $O_2$/5% $CO_2$) was perfused over the slices at ~2 mL/min throughout recordings. Dorsal striatal SPNs were visualized under IR-DIC optics (Zeiss Axioskop2; Oberkocken, Germany) at 40 x and D1- or D2-SPNs were confirmed by eGFP expression with brief observation in the epifluorescent channel. D1-SPNs (eGFP-positive in D1-eGFP mice, or eGFP-negative in D2-eGFP mice) or D2-SPNs (eGFP-positive in D2-eGFP mice, or eGFP-negative in D1-eGFP mice) that were ChR2-mCherry-negative, but in the injection site and surrounded by cells expressing ChR2-mCherry were targeted for recording. This configuration ensured that the distance between recorded and starter cells did not exceed 100 µm, maintaining close anatomical proximity and thereby preserving the likelihood of shared cortical innervation within the examined circuitry. Only animals with high-efficiency labeling throughout the cortex were used for recordings to determine collateralization.

Voltage clamp recordings were performed using 3–4 MΩ patch pipettes (WPI; Sarasota, FL), which were pulled from borosilicate glass on a P-97 pipette puller (Sutter Instruments; Novato, CA) and filled with a Cs⁺ methanesulfonate based internal solution [consisting of (in mM): $CsMeSO_3$ 120, NaCl 5, TEA-Cl 10, HEPES 10, QX-314 5 EGTA 1.1, Mg-ATP 4, Na-GTP 0.3, pH = 7.2–7.3, 305 mOsm]. All cells were voltage clamped at –70 mV during recording. Five minutes post break-in, paired light pulses (473 nm, 5–25 mW/mm², 2.5 ms, 50 ms ISI) were delivered through a glass fiber optic (200 µm in diameter, Thor Labs; Newton, NJ), positioned close to the recorded cell (50–150 µm), at 0.05 Hz using a 473 nm blue DPSS laser system (Laserglow Technologies, Toronto, ON). Light-evoked currents were collected after at least 8–10 min of bath application of 50–100 µM picrotoxin (MilliporeSigma, St. Louis, MO) to block any ChR2-mediated fast $GABA_AR$ transmission. Twenty sweeps were collected to determine latency and CV. At the end of experiments, both 10 µM NBQX (AMPAR antagonist) and 50 µM DL-APV (NMDAR antagonist; MilliporeSigma, St. Louis, MO) were applied to block AMPAR and NMDAR-mediated transmission, respectively, to confirm the EPSCs. Series resistance was initially compensated and monitored continuously throughout the experiment, and the data were rejected if the series resistance changed by more than 20% over the duration of the recording. A cell is considered

connected if it has a detectable, reliable current (20 sweeps, 0.05 Hz) with onset latency less than 10 ms post laser-on (*Klug et al., 2018*; *Smith et al., 2016*). Voltage clamp recordings were digitized at 10 kHz and filtered at 2 kHz.

For current clamp recordings of rabies-positive pyramidal neurons in the cortex, a potassium methanesulfonate-based internal solution [(in mM): KMeSO$_4$ 135, KCl 5, CaCl$_2$ 0.5, HEPES 5, EGTA 5, Mg-ATP 2, Na-GTP 0.3, (pH = 7.3, 305 mOsm)] was used. 750ms current injections (−250–200 pA) were given to test the membrane potential response of rabies-ChR2 positive pyramidal neurons, in primary motor cortex layer 5, and followed by 20 Hz or 5 Hz optogenetic stimulation to test the response of these neurons to light. Current clamp recordings were filtered and digitized at 10 kHz. All recordings were performed using a Multiclamp 700 A amplifier (Molecular Devices; Sunnyvale, CA), digitized with Digidata 1440 (Molecular Devices; Sunnyvale, CA) and collected with pClamp 9 software (Molecular Devices; Sunnyvale, CA). Data were analyzed with Clampfit 9.

## Open field

After helper viruses' injections in the striatum, animals were put back on food and allowed to recover and viral expression. They were then injected with (EnvA) SADΔG-ChR2-mCherry virus in the striatum and implanted with fiber optics in the MCC or M1 as described above (see Surgery and Viral Injection). Then animals were allowed to recover over 3 days. On the fourth day post-injection and implantation, animals went through an open field test. They were connected to fiber-optic leads (Doric) that connected to a laser through a commutator for free movement. An additional light shield was attached at the fiber optic connection to the mouse to mask the laser light output. Following habituation to the fiber optic connections in a home cage, the mice were placed in the middle of a 41 cm x 41 cm square, white and evenly illuminated open field chamber. Custom MEDPC code delivered 20 Hz or 5 Hz stimulation (473 nm blue laser, 5 mW power at connection to mouse, 10 ms pulse width) for 15 s after every 3 min and 45 s, and each animal received three to four replicates. Mice with AAV5-EF1α-DIO-ChR2(H134R)-mCherry injected bilaterally in DMS or DLS went through similar open field tests after 4 days of recovery from surgery, with optic stimulation in DMS, DLS, or M1. Video was collected for each run and analyzed in Ethovision 8.5. To analyze the open field data, the behavior was binned in 10 s bins and distance traveled during laser on period was normalized to the averaged distances during preceding 45 s just prior to stimulation onset.

## Optogenetic intracranial self-stimulation (opto-ICSS)

In opto-ICSS experiments, two different subsets of animals were used: to stimulate D1- and D2-SPNs in DMS and DLS, or to stimulate D1- or D2-SPN-projecting cortical neurons in MCC and M1, respectively. Mice that had never experienced the operant chamber were injected with virus and implanted with fiber optics using the procedure described above. From the fourth day following surgery, the mice received ICSS training for 7 consecutive days. They were attached to fiber-optic patch cords and placed in an operant chamber. Each session began with the illumination of a house light and the extension of two levers: one active (left) and one inactive (right). Every time the mouse pressed the active lever, a 20 Hz stimulation was triggered (473 nm blue laser, 5 mW power at connection to mouse, 10 ms pulse width, 1 s duration) targeting the cell bodies in MCC or M1 that project to D1- or D2-SPNs. Each session concludes after 90 min with the retraction of the levers and the house light turning off. Continuous pressing of the lever during stimulation will not lengthen the stimulation period. Pressing of the inactive lever had no consequence and was used as a control of general activity measure of non-contingent lever pressing. All protocols were custom written in MEDPC (Med Associates).

## Sequence training and optogenetic stimulation

Prior to the injection of the rabies virus, animals were pre-trained for three weeks in fixed ratio 8 (FR8) or fixed ratio 4 (FR4) task (*Jin and Costa, 2010*; *Jin et al., 2014*). Briefly, animals were food-restricted (30 hr) to start training and weighed daily to monitor their bodyweight. They were fed approximately 2–2.5 g regular chow/mouse/day after each behavioral training session concluded to maintain around 85% of their initial weight. Animals were trained in operant chambers (21.6 cm (L) x 17.8 cm (W) x 12.7 cm (H)) housed in a sound-attenuating box (Med-Associates, St. Albans, VT) with two retractable levers to the left and right of a central food magazine and a house light (3 W, 24 V) opposite to the levers and magazine. Sucrose solution (15 µl, 10%) was delivered by a syringe pump into a metal bowl

as a reinforcer. Magazine entries were recorded using an infrared beam break detector. Behavioral chambers were controlled by MED-PC IV software (MED Associates, VT) that recorded all timestamps of lever presses and magazine entries with a resolution of 10 ms.

Operant training began with continuous reinforcement (CRF) also known as fixed ratio 1 (FR1) in which animals received a reinforcer following each lever press. The animals were trained on CRF for both levers (separate flanking sessions) over 3 days, and the order of lever presentation was counterbalanced. Each session began with the illumination of the house light and the extension of one lever. The session ended with the offset of the house light and retraction of the lever after 90 min of training or after a reinforcer cap was reached. On days 1, 2, and 3, the mice could earn up to 10, 15, or 30 sucrose reinforcers, respectively. After the animals acquired CRF over 3 days, they were transitioned to FR4 and FR8 schedules on independent levers, and the order was counterbalanced. The session began with the illumination of the house light and the extension of either the left or right lever. Following four consecutive lever presses (FR4), mice received a reinforcer in a central magazine port. There was no time requirement for completion of the action sequence. The session concluded with the retraction of the lever and the offset of the house light after the mouse received either 80 reinforcers or 90 min expired. Another session was given just following the conclusion of the FR4 session, where eight consecutive lever presses (FR8) on the opposite lever resulted in the delivery of a sucrose reinforcer. The order of training FR4 or FR8 was randomly shuffled over 21 days pre-training. Left and right levers were randomly assigned FR4 or FR8 schedules, and that setup was maintained for each animal over pre-training.

On the fourth day after rabies injection and fiber optic implant, and after open field test, the mice were food deprived for 24 hr to start optogenetic test in sequence tasks. On the fifth day, the animals were tethered to two fiber-optic patch cables attached to a commutator (Doric, Canada) to allow for free rotation and placed back in the original pre-training operant box. They were given 3 days of re-training in a session of FR4 on one lever and a subsequent session of FR8 on the opposite lever with fiber attached (90 min session, 80 reinforcers max). The order of the sessions was randomly shuffled. If the animals successfully completed 80 reinforcers, they were transitioned to optogenetic stimulation test session. On day 8 post-rabies injection, optogenetic stimulations (20 Hz, 473 nm blue laser, 5 mW power at connection to mouse, 10 ms pulse width) were randomly delivered for 8 s (a time period covering roughly the entire lever press sequence) on the first press (defined by the first lever press after either head entry or 2 s break after the reward delivery) with a 50% likelihood of control non-stimulated trials randomly interleaved (*Geddes et al., 2018*). Stimulus conditions were repeated on multiple days if needed to collect enough trials for statistics. On day 12 post-rabies injection, the animals were perfused for histology analysis.

All sequence data were analyzed in MATLAB using custom scripts. To construct the peri-event time histograms (PETH), all lever presses before the reward (control or stimulation trials) were aligned to the first press of the FR4 or FR8 sequence, averaged in 100 ms bins, and filtered with a Gaussian low-pass filter (window size = 5, standard deviation = 5). All the PETHs were plotted with the first press omitted for illustration and comparison clarity. The effects of optogenetic modulation on press rate were qualitatively similar for FR4 and FR8 sequences and thus combined for statistics.

## Histology and microscopy

Approximately twelve days following rabies injection or after behavior tests, mice were anesthetized with an overdose of ketamine/xylazine and transcardially perfused with 0.01 M PBS (30–40 mL) followed by 4% paraformaldehyde (PFA)/0.1 M PB, pH 7.4 (30–40 mL), with a peristaltic perfusion pump (Cole Parmer, Vernon Hills, IL; *Klug et al., 2018*; *Smith et al., 2016*). The brain was carefully extracted and post-fixed in 4% PFA/0.1 M PB overnight (16–24 hr), then transferred to 30% sucrose/0.1 M PB for 1–2 days until the brain equilibrated and sunk. On the day of cutting, it was coronally blocked with a brain matrix (Zivic Instruments; Pittsburg, PA) and mounted on a freezing microtome. Coronal slices were collected from the most rostral to the most caudal sites at 50 μm resolution in 96 well plates containing cryoprotectant (0.1 M phosphate buffer, ethylene glycol, glycerol) to maintain AP position. Brain slices surrounding the injection site and fiber implant site were mounted on super frost plus slides (Thermo Fisher Scientific, Waltham, MA), counterstained for DAPI and cover slipped with Aqua-Poly/Mount mounting media (Polysciences, Inc; Warrington, PA). Slides

were scanned on an automated slide scanner (Olympus VS120) at 10 x in the blue and red channels. Images were batch converted to composite TIFFs and saved for image analysis.

## Statistics

Statistics were conducted in Graph Pad Prism 6.01 (La Jolla, CA). Fisher's exact test was used in comparing the likelihood of connections in slice recordings. Student unpaired two-tailed t-test was used in open field test and sequence operant task to analyze optogenetic stimulation effects. Non-parametric Mann-Whitney U Test was conducted when distributions significantly deviated from normal distributions. The reinforcing effect of each group in the ICSS task was assessed using a permutation test (10,000 permutations per group). A one-sample two-tailed t-test or Wilcoxon signed-rank test was used to assess the effectiveness of optogenetic activation in each individual group during the sequence operant task.

## Acknowledgements

The authors would like to thank Tom Jessell, Chris Kintner and members of the Jin lab for discussion and comments on the manuscript. This work was supported by grants from the NIH (R01NS083815), the Dystonia Medical Research Foundation and the McKnight Memory and Cognitive Disorders Award to XJ.

## Additional information

### Funding

| Funder | Grant reference number | Author |
|---|---|---|
| National Institutes of Health | R01NS083815 | Xin Jin |
| Dystonia Medical Research Foundation | | Xin Jin |
| McKnight Foundation | McKnight Endowment Fund for Neuroscience | Xin Jin |

The funders had no role in study design, data collection and interpretation, or the decision to submit the work for publication.

### Author contributions

Jason R Klug, Conceptualization, Data curation, Formal analysis, Validation, Investigation, Visualization, Methodology, Writing – original draft; Xunyi Yan, Data curation, Formal analysis, Validation, Investigation, Visualization, Methodology, Writing – original draft, Writing – review and editing; Hilary Hoffman, Methodology; Max D Engelhardt, Data curation, Methodology; Fumitaka Osakada, Edward M Callaway, Resources, Methodology; Xin Jin, Conceptualization, Resources, Formal analysis, Supervision, Funding acquisition, Validation, Investigation, Writing – original draft, Project administration, Writing – review and editing

### Author ORCIDs

Xunyi Yan https://orcid.org/0009-0008-7669-077X
Edward M Callaway https://orcid.org/0000-0002-6366-5267
Xin Jin https://orcid.org/0000-0002-1106-4013

### Ethics

All procedures were approved by the Salk Institute Institutional Animal Care and Use Committee (IACUC; protocol 12-00032) and conducted in accordance with NIH guidelines for the care and use of laboratory animals. For G-deleted rabies-mediated retrograde tracing and slice electrophysiology experiments, all surgeries were performed under aseptic conditions with animals anesthetized using ketamine (100 mg/kg) and xylazine (10 mg/kg). For optogenetic behavioral experiments assessing D1- or D2-SPN-projecting cortical neurons, animals were anesthetized with

isoflurane (4% induction, 1-2% maintenance). In all procedures, every effort was made to minimize animal suffering.

Joint Public Review: https://doi.org/10.7554/eLife.92992.4.sa1
Author response https://doi.org/10.7554/eLife.92992.4.sa2

---

# Additional files

## Supplementary files
MDAR checklist

## Data availability
All data generated or analysed during this study are included in the manuscript and supporting files.

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
