## [Editor Report · eLife Assessment]

This manuscript presents an **important** finding that D1- and D2-striatal neurons receive distinct cortical inputs, offering key insights into corticostriatal function. For instance, in the context of striatal-dependent learning, this distinction is highly informative for interpreting synaptic physiology data, particularly when inputs to one neuron subtype may change independently of the other. The strength of the evidence is **solid**, with anatomical and electrophysiological findings aligning well with results from optogenetic and behavioral studies. The study would be of interest to neuroscientists studying basal ganglia circuits in health and disease.

---

## [Referee Report · Joint Public Review]

Summary:

Klug et al. use monosynaptic rabies tracing of inputs to D1- vs D2-SPNs in the striatum to study how separate populations of cortical neurons project to D1- and D2-SPNs. They use rabies to express ChR2, then patch D1-or D2-SPNs to measure synaptic input. They report that cortical neurons labeled as D1-SPN-projecting preferentially project to D1-SPNs over D2-SPNs. In contrast, cortical neurons labeled as D2-SPN-projecting project equally to D1- and D2-SPNs. They go on to conduct pathway-specific behavioral stimulation experiments. They compare direct optogenetic stimulation of D1- or D2-SPNs to stimulation of MCC inputs to DMS and M1 inputs to DLS. In three different behavioral assays (open field, intra-cranial self-stimulation, and a fixed ratio 8 task), they show that stimulating MCC or M1 cortical inputs to D1-SPNs is similar to D1-SPN stimulation, but that stimulating MCC or M1 cortical inputs to D2-SPNs does not recapitulate the effects of D2-SPN stimulation (presumably because both D1- and D2-SPNs are being activated by these cortical inputs).

Strengths:

Showing these same effects in three distinct behaviors is strong. Overall, the functional verification of the consequences of the anatomy is very nice to see. It is a good choice to patch only from mCherry-negative non-starter cells in the striatum. This study adds to our understanding of the logic of corticostriatal connections, suggesting a previously unappreciated structure.

Editors' note:

The concerns raised by Reviewers #1, and #2, have been addressed during the first round of revision. The specific concern raised by Reviewer #3 is about the Rabis virus-based circuit tracing itself. This version of the work has been assessed by the editors without going back to the reviewers.

---

## [Author Response]

The following is the authors’ response to the previous reviews

**Reviewer #1 (Public review):**
Summary:The study by Klug et al. investigated the pathway specificity of corticostriatal projections, focusing on two cortical regions. Using a G-deleted rabies system in D1-Cre and A2a-Cre mice to retrogradely deliver channelrhodopsin to cortical inputs, the authors found that M1 and MCC inputs to direct and indirect pathway spiny projection neurons (SPNs) are both partially segregated and asymmetrically overlapping. In general, corticostriatal inputs that target indirect pathway SPNs are likely to also target direct pathway SPNs, while inputs targeting direct pathway SPNs are less likely to also target indirect pathway SPNs. Such asymmetric overlap of corticostriatal inputs has important implications for how the cortex itself may determine striatal output. Indeed, the authors provide behavioral evidence that optogenetic activation of M1 or MCC cortical neurons that send axons to either direct or indirect pathway SPNs can have opposite effects on locomotion and different effects on action sequence execution. The conclusions of this study add to our understanding of how cortical activity may influence striatal output and offer important new clues about basal ganglia function.The conceptual conclusions of the manuscript are supported by the data, but the details of the magnitude of afferent overlap and causal role of asymmetric corticostriatal inputs on some behavioral outcomes may be a bit overstated given technical limitations of the experiments.For example, after virally labeling either direct pathway (D1) or indirect pathway (D2) SPNs to optogenetically tag pathway-specific cortical inputs, the authors report that a much larger number of "non-starter" D2-SPNs from D2-SPN labeled mice responded to optogenetic stimulation in slices than "non-starter" D1 SPNs from D1-SPN labeled mice did. Without knowing the relative number of D1 or D2 SPN starters used to label cortical inputs, it is difficult to interpret the exact meaning of the lower number of responsive D2-SPNs in D1 labeled mice (where only ~63% of D1-SPNs themselves respond) compared to the relatively higher number of responsive D1-SPNs (and D2-SPNs) in D2 labeled mice. While relative differences in connectivity certainly suggest that some amount of asymmetric overlap of inputs exists, differences in infection efficiency and ensuing differences in detection sensitivity in slice experiments make determining the degree of asymmetry problematic.It is also unclear if retrograde labeling of D1-SPN- vs D2-SPN- targeting afferents labels the same densities of cortical neurons. This gets to the point of specificity in some of the behavioral experiments. If the target-based labeling strategies used to introduce channelrhodopsin into specific SPN afferents label significantly different numbers of cortical neurons, might the difference in the relative numbers of optogenetically activated cortical neurons itself lead to behavioral differences?

We thank the reviewer for the comments and for raising additional interpretations of our results. We agree that determining the relative number of D1- versus D2-SPN starter cells would allow a more accurate estimate of connectivity. However, due to current technical limitations, achieving this level of precision remains challenging. As the reviewer also noted, differences in the number of cortical neurons targeting D1- versus D2-SPNs could introduce additional complexity to the functional effects observed in the behavioral experiments. Moreover, functional heterogeneity is likely to exist not only among cortical neurons projecting to striatal D1- or D2-SPNs, but also within the striatal D1- and D2-SPN populations themselves. Addressing these questions at the single-neuron level will require more refined viral tools in combination with improved recording and manipulation techniques. Despite these limitations, our results suggest that a subpopulation of cortical neurons selectively targets striatal D1-SPNs, supporting a functional dichotomy of pathway-specific corticostriatal subcircuits in the control of behavior.

**Reviewer #2 (Public review):**
Summary:Klug et al. use monosynaptic rabies tracing of inputs to D1- vs D2-SPNs in the striatum to study how separate populations of cortical neurons project to D1- and D2-SPNs. They use rabies to express ChR2, then patch D1-or D2-SPNs to measure synaptic input. They report that cortical neurons labeled as D1-SPN-projecting preferentially project to D1-SPNs over D2-SPNs. In contrast, cortical neurons labeled as D2-SPN-projecting project equally to D1- and D2-SPNs. They go on to conduct pathway-specific behavioral stimulation experiments. They compare direct optogenetic stimulation of D1- or D2-SPNs to stimulation of MCC inputs to DMS and M1 inputs to DLS. In three different behavioral assays (open field, intra-cranial self-stimulation, and a fixed ratio 8 task), they show that stimulating MCC or M1 cortical inputs to D1-SPNs is similar to D1-SPN stimulation, but that stimulating MCC or M1 cortical inputs to D2-SPNs does not recapitulate the effects of D2-SPN stimulation (presumably because both D1- and D2-SPNs are being activated by these cortical inputs).Strengths:Showing these same effects in three distinct behaviors is strong. Overall, the functional verification of the consequences of the anatomy is very nice to see. It is a good choice to patch only from mCherry-negative non-starter cells in the striatum. This study adds to our understanding of the logic of corticostriatal connections, suggesting a previously unappreciated structure.Weaknesses:One limitation is that all inputs to SPNs are expressing ChR2, so they cannot distinguish between different cortical subregions during patching experiments. Their results could arise because the same innervation patterns are repeated in many cortical subregions or because some subregions have preferential D1-SPN input while others do not.

Thank you for raising this thoughtful concern. It is indeed not feasible to restrict ChR2 expression to a specific cortical region using the first-generation rabies-ChR2 system alone. A more refined approach would involve injecting Cre-dependent TVA and RG into the striatum of D1- or A2A-Cre mice, followed by rabies-Flp infection. Subsequently, a Flp-dependent ChR2 virus could be injected into the MCC or M1 to selectively label D1- or D2-projecting cortical neurons. This strategy would allow for more precise targeting and address many of the current limitations.

However, a significant challenge lies in the cytotoxicity associated with rabies virus infection. Neuronal health begins to deteriorate substantially around 10 days post-infection, which provides an insufficient window for robust Flp-dependent ChR2 expression. We have tested several new rabies virus variants with extended survival times (Chatterjee et al., 2018; Jin et al., 2024), but unfortunately, they did not perform effectively or suitably in the corticostriatal systems we examined.

In our experimental design, the aim is to delineate the connectivity probabilities to D1 or D2-SPNs from cortical neurons. Our hypothesis considered includes the possibility that similar innervation patterns could occur across multiple cortical subregions, or that some subregions might show preferential input to D1-SPNs while others do not, or a combination of both scenarios. This leads us to perform a series behavior test that using optogenetic activation of the D1- or D2-projecting cortical populations to see which could be the case.

In the cortical areas we examined, MCC and M1, during behavioral testing, there is consistency with our electrophysiological results. Specifically, when we stimulated the D1-projecting cortical neurons either in MCC or in M1, mice exhibited facilitated local motion in open field test, which is the same to the activation of D1 SPNs in the striatum along (MCC: Fig 3C & D vs. I; M1: Fig 3F & G vs. L). Conversely, stimulation of D2-projecting MCC or M1 cortical neurons resulted in behavioral effects that appeared to combine characteristics of both D1- and D2-SPNs activation in the striatum (MCC: Fig 3C & D vs. J; M1: Fig 3F & G vs. M). The similar results were observed in the ICSS test. Our interpretation of these results is that the activation of D1-projecting neurons in the cortex induces behavior changes akin to D1 neuron activation, while activation of D2-projecting neurons in the cortex leads to a combined effect of both D1 and D2 neuron activation. This suggests that at least some cortical regions, the ones we tested, follow the hypothesis we proposed.

There are also some caveats with respect to the efficacy of rabies tracing. Although they only patch non-starter cells in the striatum, only 63% of D1-SPNs receive input from D1-SPN-projecting cortical neurons. It's hard to say whether this is "high" or "low," but one question is how far from the starter cell region they are patching. Without this spatial indication of where the cells that are being patched are relative to the starter population, it is difficult to interpret if the cells being patched are receiving cortical inputs from the same neurons that are projecting to the starter population. The authors indicate they are patching from mCherry-negative neurons within the region of the mCherry-positive neurons, but since the mCherry population will include both true starter cells and monosynaptically connected cells, this is not perfectly precise. Convergence of cortical inputs onto SPNs may vary with distance from the starter cell region quite dramatically, as other mapping studies of corticostriatal inputs have shown specialized local input regions can be defined based on cortical input patterns (Hintiryan et al., Nat Neurosci, 2016, Hunnicutt et al., eLife 2016, Peters et al., Nature, 2021).

This is a valid concern regarding anatomical studies. Investigating cortico-striatal connectivity at the single-cell level remains technically challenging due to current methodological limitations. At present, we rely on rabies virus-mediated trans-synaptic retrograde tracing to identify D1- or D2-projecting cortical populations. This anatomical approach is coupled with ex vivo slice electrophysiology to assess the functional connectivity between these projection-defined cortical neurons and striatal SPNs. This enables us to quantify connection ratios, for example, the proportion of D1-projecting cortical neurons that functionally synapse onto non-starter D1-SPNs.

To ensure the robustness of our conclusions, it is essential that both the starter cells and the recorded non-starter SPNs receive comparable topographical input from the cortex and other brain regions. Therefore, we carefully designed our experiments so that all recorded cells were located within the injection site, were mCherry-negative (i.e., non-starter cells), and were surrounded by ChR2-mCherry-positive neurons. This configuration ensured that the distance between recorded and starter cells did not exceed 100 µm, maintaining close anatomical proximity and thereby preserving the likelihood of shared cortical innervation within the examined circuitry.

These methodological details are also described in the section on ex vivo brain slice electrophysiology, specifically in the Methods section, lines 453–459:

“D1-SPNs (eGFP-positive in D1-eGFP mice, or eGFP-negative in D2-eGFP mice) or D2-SPNs (eGFP-positive in D2-eGFP mice, or eGFP-negative in D1-eGFP mice) that were ChR2-mCherry-negative, but in the injection site and surrounded by cells expressing ChR2-mCherry were targeted for recording. This configuration ensured that the distance between recorded and starter cells did not exceed 100 µm, maintaining close anatomical proximity and thereby preserving the likelihood of shared cortical innervation within the examined circuitry.”

This experimental strategy was implemented to control for potential spatial biases and to enhance the interpretability of our connectivity measurements.

A caveat for the optogenetic behavioral experiments is that these optogenetic experiments did not include fluorophore-only controls, although a different control (with light delivered in M1) is provided in Supplementary Figure 3. Another point of confusion is that other studies (Cui et al, J Neurosci, 2021) have reported that stimulation of D1-SPNs in DLS inhibits rather than promotes movement. This study may have given different results due to subtly different experimental parameters, including fiber optic placement and NA.

We appreciate the reviewer’s thoughtful evaluation and comments. We have added a short discussion of Cui et al.’s study on optogenetic stimulation of D1-SPNs in the DLS (lines 341-343), which reports findings that contrast with ours and those of other studies.

**Reviewer #3 (Public review):**
Review of resubmission: The authors provided a response to the reviews from myself and other reviewers. While some points were made satisfactorily, particularly in clarification of the innervation of cortex to striatum and the effects of input stimulation, many of my points remain unaddressed. In several cases, the authors chose to explain their rationale rather than address the issues at hand. A number of these issues (in fact, the majority) could be addressed simply by toning done the confidence in conclusions, so it was disappointing to see that the authors by and large did not do this. I repeat my concerns below and note whether I find them to have been satisfactorily addressed or not.In the manuscript by Klug and colleagues, the investigators use a rabies virus-based methodology to explore potential differences in connectivity from cortical inputs to the dorsal striatum. They report that the connectivity from cortical inputs onto D1 and D2 MSNs differs in terms of their projections onto the opposing cell type, and use these data to infer that there are differences in cross-talk between cortical cells that project to D1 vs. D2 MSNs. Overall, this manuscript adds to the overall body of work indicating that there are differential functions of different striatal pathways which likely arise at least in part by differences in connectivity that have been difficult to resolve due to difficulty in isolating pathways within striatal connectivity, and several interesting and provocative observations were reported. Several different methodologies are used, with partially convergent results, to support their main points.However, I have significant technical concerns about the manuscript as presented that make it difficult for me to interpret the results of the experiments. My comments are below.Major:There is generally a large caveat to the rabies studies performed here, which is that both TVA and the ChR2-expressing rabies virus have the same fluorophore. It is thus essentially impossible to determine how many starter cells there are, what the efficiency of tracing is, and which part of the striatum is being sampled in any given experiment. This is a major caveat given the spatial topography of the cortico-striatal projections. Furthermore, the authors make a point in the introduction about previous studies not having explored absolute numbers of inputs, yet this is not at all controlled in this study. It could be that their rabies virus simply replicates better in D1-MSNs than D2-MSNs. No quantifications are done, and these possibilities do not appear to have been considered. Without a greater standardization of the rabies experiments across conditions, it is difficult to interpret the results.This is still an issue. The authors point out why they chose various vectors. I can understand why the authors chose the fluorophores etc. that they did, yet the issues I raised previously are still valid. The discussion should mention that this is a potential issue. It does not necessarily invalidate results, but it is an issue. Furthermore, it is possible (in all systems) that rabies replicates better/more efficiently in some cells than others. This is one possible interpretation that has not really been explored in any study. I don't suggest the authors attempt to do that, but it should be raised as a potential interpretation. If the rabies results could mean several different things, the authors owe it to the readership to state all possible interpretations of data.

We thank the reviewer for the comments and suggestions. Because the same fluorophore (mCherry) was used in both TVA- and ChR2-expressing viruses, it was not possible to distinguish true starter SPNs from TVA-only SPNs or monosynaptically labeled SPNs. This limitation makes it difficult to precisely assess the efficiency of rabies labeling and retrograde tracing in our experimental setup. Moreover, differences in rabies replication efficiency between D1- and D2-SPNs could potentially lead to an apparent lower connection probability from D1-projecting cortical neurons to D2-SPNs than from D2-projecting cortical neurons to D1-SPNs. We have added this clarification to the Discussion (lines 280-297).

The authors claim using a few current clamp optical stimulation experiments that the cortical cells are healthy, but this result was far from comprehensive. For example, membrane resistance, capacitance, general excitability curves, etc are not reported. In Figure S2, some of the conditions look quite different (e.g., S2B, input D2-record D2, the method used yields quite different results that the authors write off as not different). Furthermore, these experiments do not consider the likely sickness and death that occurs in starter cells, as has been reported elsewhere. Health of cells in the circuit is overall a substantial concern that alone could invalidate a large portion, if not all, of the behavioral results. This is a major confound given those neurons are thought to play critical roles in the behaviors being studied. This is a major reason why first-generation rabies viruses have not been used in combination with behavior, but this significant caveat does not appear to have been considered, and controls e.g., uninfected animals, infected with AAV helpers, etc, were not included.This issue remains unaddressed. I did not request clarity about experimental design, but rather, raised issues about the potential effects of toxicity. I believe this to be a valid concern that needs to be discussed in the manuscript, especially given what look visually like potential differences in S2.

We understand and appreciate the reviewer’s concern regarding the potential cytotoxicity of rabies virus infection. Although we performed the in vivo optogenetic behavioral experiments during a period when rabies-infected cells are generally considered relatively healthy, some deficits in starter cells may still occur and could contribute to the observed effects of optogenetic cortical stimulation. We have added this clarification to the Discussion (lines 298-306).

The overall purity (e.g., EnvA pseudotyping efficiency) of the RABV prep is not shown. If there was a virus that was not well EnvA-pseudotyped and thus could directly infect cortical (or other) inputs, it would degrade specificity. This issue has not been addressed. Viral strain is irrelevant. The quality of the specific preparations used is what matters.While most of the study focuses on the cortical inputs, in slice recordings, inputs from the thalamus are not considered, yet likely contribute to the observed results. Related to this, in in vivo optogenetic experiments, technically, if the thalamic or other inputs to the dorsal striatum project to the cortex, their method will not only target cortical neurons but also terminals of other excitatory inputs. If this cannot be ruled it, stating that the authors are able to selectively activate the cortical inputs to one or the other population should be toned down.The authors added text to the discussion to address this point. While it largely does what is intended, based on the one study cited, I disagree with the authors' conclusions that it is "clear" that potential contamination from other sites does not play a role. The simplest interpretation is the one the authors state, and there is some supporting evidence to back up that assertion, but to me that falls short of making the point "clear" that there are no other interpretations.The statements about specificity of connectivity are not well founded. It may be that in the specific case where they are assessing outside of the area of injections, their conclusions may hold (e.g., excitatory inputs onto D2s have more inputs onto D1s than vice versa). However, how this relates to the actual site of injection is not clear. At face value, if such a connectivity exists, it would suggest that D1-MSNs receive substantially more overall excitatory inputs than D2s. It is thus possible that this observation would not hold over other spatial intervals. This was not explored and thus the conclusions are over-generalized. e.g., the distance from the area of red cells in the striatum to recordings was not quantified, what constituted a high level of cortical labeling was not quantified, etc. Without more rigorous quantification of what was being done, it is difficult to interpret the results.Again, the goal here would be to make a statement about this in the discussion to clarify limitations of the study. I don't expect the authors to re-do all of these experiments, but since they are discussing the corticostriatal circuits, which have multiple subdomains, this remains a relevant point. It has not been addressed.The results in Figure 3 are not well controlled. The authors show contrasting effects of optogenetic stimulation of D1-MSNs and D2-MSNs in the DMS and DLS, results which are largely consistent with the canon of basal ganglia function. However, when stimulating cortical inputs, stimulating the inputs from D1-MSNs gives the expected results (increased locomotion) while stimulating putative inputs to D2-MSNs had no effect. This is not the same as showing a decrease in locomotion - showing no effect here is not possible to interpret.I think that the caveat of showing no clear effects of inputs to D2 stimulation should be pointed out. Yes, I understand that the viruses appeared to express etc., but again it remains possible that the results are driven by a lack of e.g., sufficient ChR2 expression. Aside from a full quantification of the number of cells expressing ChR2, overlap in fiber placement and ChR2 expression (which I don't suggest), this remains a possibility and should be pointed out, as it remains a possibility.In the light of their circuit model, the result showing that inputs to D2-MSNs drive ICSS is confusing. How can the authors account for the fact that these cells are not locomotor-activating, stimulation of their putative downstream cells (D2-MSNs) does not drive ICSS, yet the cortical inputs drive ICSS? Is the idea that these inputs somehow also drive D1s? If this is the case, how do D2s get activated, if all of the cortical inputs tested net activate D1s and not D2s? Same with the results in Figure 4 - the inputs and putative downstream cells do not have the same effects. Given potential caveats of differences in viral efficiency, spatial location of injections, and cellular toxicity, I cannot interpret these experiments.The explanation the authors provide in their rebuttal makes sense, however this should be included in the discussion of the manuscript, as it is interesting and relevant.

We thank the reviewer for the valuable comments and suggestions. In line with the reviewer’s recommendation, we have incorporated these explanations into the Discussion (lines 242–279) to help interpret the complex behavioral outcomes of optogenetic stimulation of cortical neurons projecting to D1- or D2-SPNs.

**Reviewer #2 (Recommendations for the authors):**
I appreciate the authors' responses, which helped clarify some experimental choices. I appreciate that the experiment in Fig S3 serves as a reasonable light control for optogenetics experiments. The careful comparison with methods in Cui et al (2021) is useful, although not added to the main manuscript. Some of the other citations here don't really address the controversy, e.g. Kravitz at al is in DMS, but perhaps fully addressing this issue is outside the scope of the current manuscript and awaits further experiments. I also appreciate the clarification for recording locations that "This configuration ensured that the distance between recorded and starter cells did not exceed 100 µm, maintaining close anatomical proximity and thereby preserving the likelihood of shared cortical innervation within the examined circuitry." However, the statement in the reviewer response does not seem to be added to the manuscript's methods, which I think would be helpful. The criteria for choosing recorded cells are still a bit fuzzy without a map of recording locations and histology. There is also a problem that mCherry-positive cells could be starter cells or could be monosynaptically traced cells, so it is hard to know the area of the starter cell population in these experiments for sure. My evaluation of the manuscript remains largely the same as the original. However, I have adjusted my public review a bit to incorporate the authors' responses. I still think this paper has valuable information, suggesting an interesting and previously unappreciated structure of corticostriatal inputs that I hope this group and others will continue to investigate and incorporate into models of basal ganglia function.

We thank the reviewer for the valuable suggestions. We have now included a comparison with Cui et al. in the Discussion. In addition, we have added the criteria for selecting recorded cells to the Methods section: ‘This configuration ensured that the distance between recorded and starter cells did not exceed 100 µm, maintaining close anatomical proximity and thereby preserving the likelihood of shared cortical innervation within the examined circuitry.’